# Enhancing Tactile-based Reinforcement Learning for Robotic Control

**Elle Miller**     **Trevor McInroe**     **David Abel**     **Oisin Mac Aodha**     **Sethu Vijayakumar**

University of Edinburgh

## Abstract

Achieving safe, reliable real-world robotic manipulation requires agents to evolve beyond vision and incorporate tactile sensing to overcome sensory deficits and reliance on idealised state information. Despite its potential, the efficacy of tactile sensing in reinforcement learning (RL) remains inconsistent. We address this by developing self-supervised learning (SSL) methodologies to more effectively harness tactile observations, focusing on a scalable setup of proprioception and sparse binary contacts. We empirically demonstrate that sparse binary tactile signals are critical for dexterity, particularly for interactions that proprioceptive control errors do not register, such as decoupled robot-object motions. Our agents achieve superhuman dexterity in complex contact tasks (ball bouncing and Baoding ball rotation). Furthermore, we find that decoupling the SSL memory from the on-policy memory can improve performance. We release the Robot Tactile Olympiad (`RoTO`) benchmark to standardise and promote future research in tactile-based manipulation. *Project page:* `https://elle-miller.github.io/tactile_rl`

## 1   Introduction

Imagine a humanoid robot gently lifting an elderly person out of their bed with the same delicacy as a human caregiver, or a person with severe motor impairments using a re-enabling robotic arm to brush their teeth. These scenarios represent critical pathways toward enhancing human independence, dignity, and quality of life. For this future to safely materialise, robots must evolve beyond merely *seeing* their environment; they must possess some capacity to *feel* it.

Due to the scalability challenges associated with collecting dexterous demonstrations with tactile data, reinforcement learning (RL) is a primary candidate for enabling tactile-based manipulation. While RL has facilitated breakthroughs in locomotion, yielding robust agents that can traverse complex terrains [49], manipulation lags significantly behind. This deficit stems from multiple challenges, including inaccuracies in contact simulation, the need for complex reward functions, and critically, a heavy reliance on idealised state information (e.g., prior work required 19 cameras for object and fingertip localisation [5]). In this work, we tackle this reliance by suggesting that tactile data can effectively replace idealised state information. Given that manipulation is fundamentally defined by control through selective contact [37], we argue that tactile sensing provides a sufficiently rich information stream to localise and control objects, paving the way for more scalable robotic systems.

Despite a decade of focused interest in tactile-based RL [33, 40, 53, 61, 62, 68], the hoped-for breakthrough manipulation capabilities have yet to fully materialise. We posit that progress in this domain remains stymied by two critical, interrelated challenges. First, a lack of hardware and data convergence (sensor type, placement strategy, information representation) has hindered systematic progress. Second, the published literature provides conflicting evidence regarding the utility of tactile feedback in RL. Some studies report marginal performance gains when incorporating tactile information to the observations [40, 41], while others do not [24, 44, 62]. For instance, [45, 54]

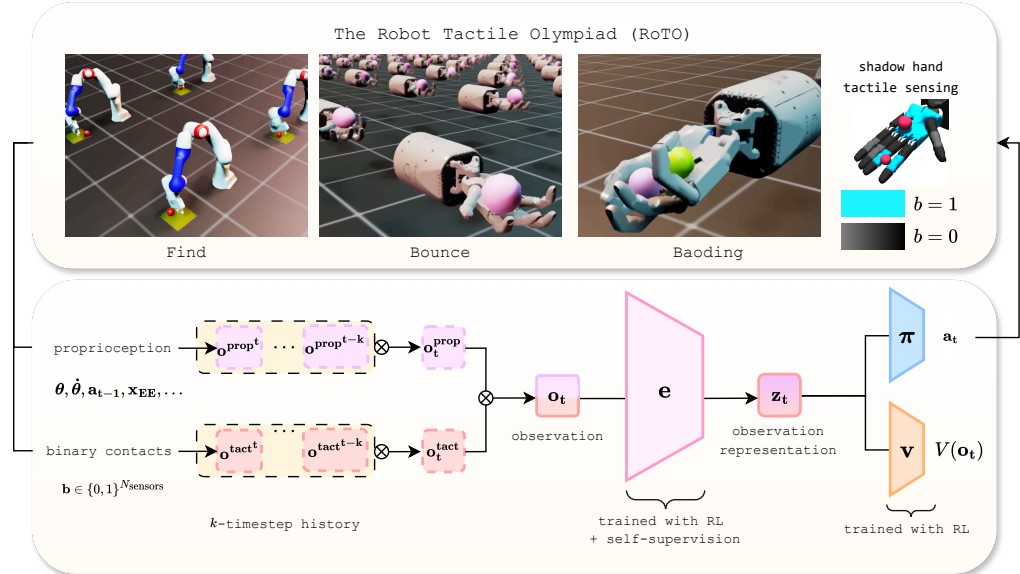

Figure 1: **Tactile-based RL with self-supervision**. Our agents achieve superhuman dexterity in our RoTO benchmark using only a history of proprioception and binary contacts (i.e., with no vision or privileged information). By jointly training the observation encoder with self-supervision, the agent transforms complex sensory input into representations that capture object positions and velocities.

achieve impressive in-hand manipulation using proprioceptive history alone, with [46] claiming that binary contacts are already implicitly contained in proprioceptive history.

We hypothesise that the reported performance discrepancies in tactile reinforcement learning arise from the unique data characteristics that tactile feedback presents to deep RL. Tactile measurements are often sparse (only present during contact) and non-smooth, potentially leading to unbalanced datasets and learning instabilities. Consequently, a deep RL agent may struggle to extract a useful representation from this raw data, causing agents to converge prematurely to suboptimal policies reliant on continuous proprioceptive signals. To alleviate the significant burden of simultaneously learning an observation representation, policy, and value function, we propose using Self-Supervised Learning (SSL) to assist with learning the observation representation [25]. While existing tactile SSL objectives have shown some utility (e.g., pixel augmentation [19, 24], masked reconstruction [53]) we posit that they often fail to encourage the encoding of critical features from state transitions, such as object velocity or friction, that are essential for more complex control.

Therefore, the central objective of this work is to develop general-purpose SSL methodologies that effectively leverage tactile observations across diverse robotic control tasks in RL. To harness the power of tactile sensing and avoid interference from complementary modalities, we deliberately do not study pixel or depth observations. Furthermore, to maximise future utility and real-world scalability, we focus on the simplest, lowest-cost tactile setup: *sparse binary contacts* (e.g., Figure 1), which conveniently avoids the sim-to-real challenges associated with continuous measurements [71]. Finally, guided by the intuition that robust blind manipulation should be possible, we forgo the traditional reliance on teacher-student networks and instead focus on learning powerful representations directly from the sensory information using self-supervision. Our approach proves highly effective, demonstrating that direct sensory learning provides the necessary control capacity without ever necessitating idealised state information. Our main contributions are as follows:

- We provide **empirical evidence that sparse binary tactile signals are critical for dexterity**, showing they can significantly improve performance beyond what is achievable with proprioceptive history alone. We propose that explicit tactile information is essential only where proprioceptive joint control errors fail to reliably register all environment dynamics, e.g., decoupled object-robot dynamics, low-inertia objects, contact spatial ambiguity, and multi-contact resolution.
- We demonstrate a **new level of simulated superhuman dexterity in complex contact tasks** using a limited sensory setup of only proprioception and 17 binary contacts.

- **We propose and analyse four SSL objectives** for tactile agents (tactile reconstruction, full reconstruction, forward dynamics, and tactile forward dynamics), demonstrating superior performance to RL-only approaches. We find that forward dynamics is the most effective objective, creating representations that encode object positions and velocities.
- We provide **empirical evidence that decoupling the SSL training data from the on-policy memory can improve performance**, demonstrating possibilities for leveraging off-policy experience.
- **We introduce the** `Robot Tactile Olympiad (RoTO)` **benchmark**, comprising three challenging Isaac Lab environments (*Find*, *Bounce*, and *Baoding*), with tuned baselines and integrated hyperparameter optimisation to standardise and inspire future research in tactile manipulation.

## 2 Related work

**In-hand manipulation with reinforcement learning (RL).** Most prior work on in-hand object manipulation relies on either privileged information with teacher-student networks [45, 46, 69], visual input (RGB-D) [8, 46, 73], or explicit object pose estimators [23, 43, 57, 67]. For Baoding ball rotation, [43] use tracking cameras for ball positions, and [73] use pointclouds with privileged information distillation. To our knowledge, the most advanced dexterous 'blind' RL demos are object rotation [54, 71] and half-rotation of Baoding balls [67]. In terms of speed, the fastest robot policies in both simulation and the real world achieve a maximum of 3 complete rotations in 10 seconds [73]. In contrast, our blind simulated agents achieve up to 25 rotations over the same time interval.

**Pixel representation learning in RL.** RL agents often struggle to learn effective representations from high-dimensional inputs like pixels. While early pixel-based RL methods did not specifically target this problem, most contemporary work leverages auxiliary objectives to guide representation learning [25]. Reconstruction is a common objective, explored in both model-free [28, 29, 70] and model-based paradigms [20, 21, 22, 32, 63, 66]. However, achieving pixel-perfect reconstruction can retain information irrelevant to control [55]. Consequently, many modern auxiliary objectives operate in the latent space, including forward dynamics [16, 39, 51], disentanglement [11, 12, 13], contrastive approaches [1, 31, 56], and information-theoretic objectives [34, 38]. In this work, we examine how these techniques can be applied to the proprioceptive and tactile robotic sensory modalities.

**Multimodal representation learning in RL.** Recent works have aimed to produce a generalisable auxiliary objectives that operate over all modalities. [7] propose maximising mutual information (MI) between unimodal and multimodal representations to align the latent spaces. Noting this does not filter irrelevant information, [72] suggest using an information bottleneck [59] that maximises forward dynamics information. [6] propose that self-supervised objectives should be modality-specific, e.g., reconstruction for proprioception and contrastive losses for images. We do not propose a generalisable multimodal auxiliary objective, but auxiliary objectives to better leverage the tactile modality.

**Tactile representation learning in RL.** Few works leverage self-supervised objectives to improve tactile-based RL. Early work used a variational autoencoder with forward dynamics objective for a stabilisation task [60]. More recent approaches have explored pixel-based objectives like masked autoencoding [53] and augmentation [18, 24], or contrastive learning to maximise similarity between unimodal visual and tactile representations [10]. Other tactile-specific objectives include predicting the presence of contacts [9, 35]. Outside of mode-free RL, tactile dynamics models have been learned for estimating object features [64] and model predictive control (MPC) [2, 58]. Similar to these works and [60], we identify dynamics as a promising objective for tactile information and are the first to study multi-step forward dynamics with tactile data in model-free RL, focusing on binary contacts.

**Tactile-tailored RL.** Given that tactile interactions are both important and sparse, some research focuses on modifying the RL setup. Prior work has proposed increasing the sampling probability of contact-rich episodes in off-policy algorithms [62] or only updating the tactile encoder under contact [24]. To the best of our knowledge, we are the first in tactile-based on-policy RL to modify the dataset used for the self-supervised objective, deviating from the typical on-policy rollout.

**Analysing representations in RL.** Despite extensive research into representation learning in RL, relatively little work focuses on understanding the properties of the learned representations themselves [15, 65]. The most common analysis technique is linear probing, used to assess how well the latent space encodes environment features [4, 47, 74]. For our setting, we evaluate the representation by its ability to encode the true underlying environment state [36], quantifying this relationship using mutual information estimation [15].

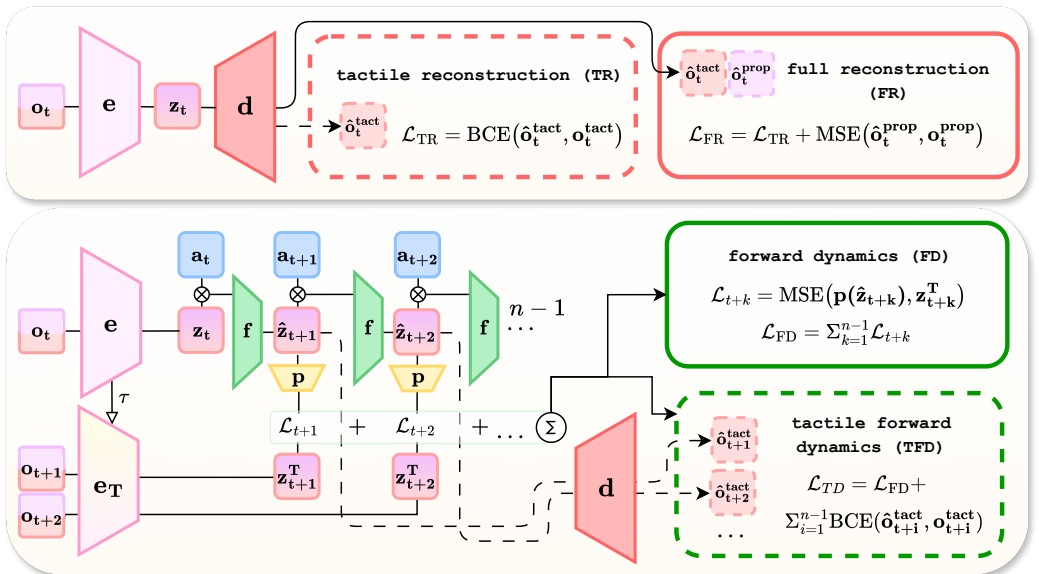

Figure 2: **Proposed self-supervised objectives.** Reconstruction losses (TR, FR) force the encoder to preserve tactile information within $\mathbf{z_t}$ by using a binary classification decoder to reconstruct the original $\mathbf{o_t^{tact}}$. Forward dynamics losses (FD, TFD) ensures the encoder extracts the necessary information from $\mathbf{o_t}$ for accurate forecasting of the representation $\mathbf{z_t}$ several timesteps into the future.

## 3 Method

### 3.1 Problem setting

We study a specialised Partially-Observable Markov Decision Process (POMDP) [26] parameterised by the tuple $\langle \mathcal{S}, \mathcal{O}, F, \mathcal{A}, T, R, \gamma \rangle$. The state space $\mathbf{s} \in \mathcal{S}$, action space $\mathbf{a} \in \mathcal{A}$, transition kernel $T : \mathcal{S} \times \mathcal{A} \times \mathcal{S} \to [0, 1]$, reward function $R : \mathcal{S} \times \mathcal{A} \to \mathbb{R}$, and discount factor $\gamma \in (0, 1)$ are standard. The true state $\mathbf{s}$ is unobservable; instead, the agent receives a composite observation $\mathbf{o} \in \mathcal{O}$, which is defined as a $k$-timestep history of multi-modal readings. The observation function $F : \mathcal{S} \to \mathcal{O}$ is deterministic, with partial observability arising from perceptual aliasing (i.e., where distinct states $\mathbf{s_1} \neq \mathbf{s_2}$ can produce the same observation $F(\mathbf{s_1}) = F(\mathbf{s_2})$ due to sensing limitations). The agent interacts via a policy $\pi : \mathcal{O} \to \mathcal{P}(\mathcal{A})$ that parameterises a distribution over actions, seeking the optimal policy $\pi^\star = \arg\max_\pi \mathbb{E}\left[\sum_{t=1}^\infty \gamma^{t-1} R(\mathbf{s_t}, \mathbf{a_t})\right]$.

We solve our POMDP using RL, implemented as follows (Figure 1). The agent receives the observation $\mathbf{o_t}$, which is a concatenation of the $k$-length history for proprioceptive ($\mathbf{o_t^{prop}}$: joint angles $\boldsymbol{\theta}$, joint velocities $\dot{\boldsymbol{\theta}}$, last action $\mathbf{a_{t-1}}$, and if applicable, the end-effector pose $\mathbf{x_{EE}}$, $\mathbf{q_{EE}}$) and tactile ($\mathbf{o_t^{tact}}$: binary contacts $\mathbf{b} \in \{0, 1\}^{N_{\text{sensors}}}$) modalities. The agent is comprised of three MLPs: an observation encoder $\mathbf{e}$, policy $\pi$, and value function $\mathbf{v}$. The observation encoder ($\mathbf{o_t} \to 1024 \to 512 \to 256 \to \mathbf{z_t}$) learns a compact representation $\mathbf{z_t}$, which $\pi$ and $\mathbf{v}$ are conditioned on ($\pi : \mathbf{z_t} \to 128 \to 64 \to \mathbf{a_t}$, $\mathbf{v} : \mathbf{z_t} \to 128 \to 64 \to V(\mathbf{o_t})$). We train using the clip variant of Proximal Policy Optimisation (PPO) [50], augmented with a self-supervised auxiliary loss $\mathcal{L}_{\text{aux}}$ for representation learning. The total loss $\mathcal{L}$ is minimised via backpropagation:

$$\mathcal{L}_{\text{PPO}}(\theta_e, \theta_\pi, \theta_v) = L_\pi^{\text{CLIP}}(\theta_e, \theta_\pi) - c_V \mathcal{L}_V(\theta_e, \theta_v) + c_{\text{ent}} \mathcal{L}_{\text{entropy}}(\theta_e, \theta_\pi), \tag{1}$$

$$\mathcal{L} = \mathcal{L}_{\text{PPO}}(\theta_e, \theta_\pi, \theta_v) + c_{\text{aux}} \mathcal{L}_{\text{aux}}(\theta_e, \theta_{aux}). \tag{2}$$

The PPO update (optimising $\mathbf{e}$, $\pi$, and $\mathbf{v}$) is performed using separate optimisers with a shared learning rate $lr$ and gradient clipping at $1.0$. The auxiliary loss uses a separate optimiser (optimising $\mathbf{e}$ and auxiliary networks) with a different learning rate $lr_{\text{aux}}$ and is applied without gradient clipping.

## 3.2 Self-supervised objectives

The self-supervised loss $\mathcal{L}_{\text{aux}}$ is computed using auxiliary networks ($\theta_{\text{aux}}$) that are discarded post-training. We propose four distinct auxiliary learning objectives, $\mathcal{L}_{\text{aux}} \in \{\mathcal{L}_{\text{TR}}, \mathcal{L}_{\text{FR}}, \mathcal{L}_{\text{FD}}, \mathcal{L}_{\text{TFD}}\}$, described below, which are used to train $\mathbf{e}$ and the auxiliary networks (architectures in Appendix F).

**Tactile Reconstruction (TR).** To mitigate the observed tendency of gradient-based optimisation to prematurely favor features from the stable proprioceptive history, thereby neglecting the complex tactile input, we introduce a reconstruction objective to exclusively decode the tactile observation $\mathbf{o}_t^{\text{tact}}$ from the learned multimodal representation $\mathbf{z}_t$ with a decoder $\mathbf{d}$ (Figure 2, top). Since the input is binary, TR is implemented as a binary classification problem using a Binary Cross-Entropy (BCE) loss on the logits. Given the inherent sparsity of binary contact data, we employ a positive weighting $\mathbf{p_c} = 10$ to penalise false negatives (missed contacts) more heavily. The objective minimises the difference between the predicted and actual observed tactile distribution:

$$\mathcal{L}_{\text{TR}} = \text{BCE}(\hat{\mathbf{o}}_t^{\text{tact}}, \mathbf{o}_t^{\text{tact}}) = -(\mathbf{p_c} \cdot \mathbf{o}_t^{\text{tact}} \cdot \log(\hat{\mathbf{o}}_t^{\text{tact}}) + (1 - \mathbf{o}_t^{\text{tact}}) \cdot \log(1 - \hat{\mathbf{o}}_t^{\text{tact}})). \quad (3)$$

**Full Reconstruction (FR).** For comparison, we implement a standard Full Reconstruction (FR) baseline, which attempts to decode both the tactile and proprioceptive observations simultaneously from $\mathbf{z}_t$. FR minimises the sum of the TR loss and a Mean Squared Error (MSE) loss for the continuous proprioceptive observation $\mathbf{o}_t^{\text{prop}}$:

$$\mathcal{L}_{\text{FR}} = \mathcal{L}_{\text{TR}} + \text{MSE}(\hat{\mathbf{o}}_t^{\text{prop}}, \mathbf{o}_t^{\text{prop}}). \quad (4)$$

**Forward Dynamics (FD).** Next, we propose multi-step forward dynamics as an promising alternate objective to distill the observations into predictive representations. This is achieved by encouraging the encoder to extract information from the current observation $\mathbf{o}_t$ that is essential for accurately forecasting the observation representation $\mathbf{z}_t$ several timesteps into the future. A sequence of length $n$ $(\mathbf{o}_t, \mathbf{a}_t, \ldots, \mathbf{o}_{t+n-1}, \mathbf{a}_{t+n-1})$ is sampled from a memory, ensuring no episode-terminating transitions are included. Given the current latent state $\mathbf{z}_t$ and action $\mathbf{a}_t$, a forward model $\mathbf{f}$ predicts the next latent state, $\hat{\mathbf{z}}_{t+1} = \mathbf{f}(\mathbf{z}_t, \mathbf{a}_t)$. This prediction is then used autoregressively to forecast the entire sequence (Figure 2, bottom). To provide a stable learning signal, the prediction is compared against a target latent state $\mathbf{z}_{t+i}^{\mathbf{T}} = \mathbf{e_T}(\mathbf{o}_{t+i})$, where $\mathbf{e_T}$ is a target encoder maintained as an exponential moving average (EMA) of the actual encoder $\mathbf{e}$. To decouple the dynamics learning from the prediction task, the prediction loss is calculated between a nonlinear projection of the predicted latent state $\mathbf{p}(\hat{\mathbf{z}}_{t+i})$ and the target latent state $\mathbf{z}_{t+i}^{\mathbf{T}}$, similar to [17]:

$$\mathcal{L}_{\text{FD}} = \sum_{i=1}^{n-1} \text{MSE}\Big(\mathbf{p}(\hat{\mathbf{z}}_{t+i}), \mathbf{z}_{t+i}^{\mathbf{T}}\Big). \quad (5)$$

**Tactile Forward Dynamics (TFD).** To ensure the learned dynamics latent space also models tactile dynamics, we introduce a novel objective. This combined loss optimises the encoder to predict future latent states ($\mathcal{L}_{\text{FD}}$) from which future tactile observations can be reconstructed (via the TR loss $\mathcal{L}_{\text{TR}}$). Specifically, the predicted latent states $\hat{\mathbf{z}}_{t+i}$ are decoded back into the next-step tactile space $\hat{\mathbf{o}}_{t+i}^{\text{tact}}$, and the resulting loss is added to the dynamics term:

$$\mathcal{L}_{\text{TD}} = \mathcal{L}_{\text{FD}} + \sum_{i=1}^{n-1} \text{BCE}(\hat{\mathbf{o}}_{t+i}^{\text{tact}}, \mathbf{o}_{t+i}^{\text{tact}}). \quad (6)$$

## 3.3 Separated auxiliary memory

When training on-policy RL with an auxiliary objective, the default is to jointly optimise these objectives on the same rollout data, and then permanently discard this data (e.g., [15, 30, 48]). We observe that the abrupt changes in training data cause small spikes in the auxiliary loss throughout training (Appendix B). We propose to separate out and increase the size of the auxiliary memory in multiples of $N_{rollouts}$ to improve performance through (i) stabilising the auxiliary updates and (ii) optimising on a wider distribution of data. For an agent using $B$ using parallel environments and a rollout length $R$, the on-policy RL memory has a shape $[B, R, ...]$. Our method simply involves storing data for the auxiliary tasks in a larger, separate buffer of size $[N_{rollouts}, B, R, ...]$.

## 4 Experimental setup

**RL.** We use a customised implementation of Proximal Policy Optimisation (PPO) [50] from SKRL [52] to incorporate observation stacking, self-supervision, and separated environments for continuous evaluation. We using 4096 parallelised environments for training and 100 for evaluation.

**Hyperparameters.** To account for the fundamental changes introduced by the self-supervision on the state representation, we conducted an individual hyperparameter sweep for every environment and method combination, aligning with best practices for RL research [14]. Each sweep comprised 20 trials using the TPE sampler with 5 startup trials. The swept hyperparameters included PPO hyperparameters (learning rate $lr$, rollout length, number of minibatches, number of learning epochs, entropy loss scale $c_{ent}$), self-supervision hyperparameters (learning rate $lr_{aux}$, loss weight $c_{aux}$), and for forward dynamics objectives, the sequence length $n$. All sweep information is in Appendix G.

**Compute.** Experiments were executed on a GPU cluster (8x NVIDIA RTX A4500s). The simulation environment (Isaac Lab) required 16GB VRAM, 32GB RAM, and 8 CPU cores. The approximate time for a single experiment, including the hyperparameter sweep ($\sim$ 50 hours) and the final five seeds ($\sim 5 \times 2$ hrs) totaled $\sim$ 60 hours. With seven experiments across three tasks, the cumulative paper compute time is $7 \times 3 \times 60 = 1{,}260$ hours.

**Environments.** We evaluate our method on three custom robotic manipulation tasks implemented within Isaac Lab [42] (Figure 1). They were designed to collectively cover a wide range of tactile interaction patterns (i.e., sparse, intermittent, and sustained), to comprehensively assess the generalisability of our proposed methods. We release our environments and baselines as an open-source benchmark, the `Robot Tactile Olympiad (RoTO)` to foster progress in tactile-based manipulation.

- *Find:* The agent must locate a fixed sphere within 20 cm $\times$ 20 cm (300 timesteps).

- *Bounce:* The agent must bounce a ball as many times as possible in 10 seconds (600 timesteps). A bounce is defined as a contact event after a period of at least 5 timesteps ($\sim$ 83ms) without contact. The ball is modelled off a typical office stress ball (70mm diameter, 30g). The theoretical maximum is 100 bounces (given the 5-timestep separation), while the human Guinness World Record corresponds to 59 bounces in 10 seconds (Appendix C).

- *Baoding:* The agent must rotate two balls (1.5 inch diameter, 55g) around each other in-hand as many times as possible within 10 seconds (600 timesteps). This task is inspired by the historical Chinese practice used for dexterity and mindfulness. For context, the fastest human demonstration we found online achieves 13 rotations in 10 seconds (Appendix C).

The physics simulation operates at 120 Hz, while the control policy executes at 60 Hz. All robots are joint position controlled. We implement *Find* using the Franka robot with $\mathbf{a_t} \in \mathbb{R}^9$, attaching $N_{\text{sensors}} = 2$ tactile sensors onto the fingers, with an observation history length $k = 16$. We implement *Bounce* and *Baoding* using the Shadow Hand with $\mathbf{a_t} \in \mathbb{R}^{20}$, specifying each link as a contact sensor $N_{\text{sensors}} = 17$ (Figure 1), with an observation history length $k = 4$. See Appendix E for full POMDP details including reward functions.

## 5 Experimental results

Figures 3-5 depict the mean evaluation return across 5 seeds, with $\pm 1$ standard deviation shaded.

**RL-only.** To establish a baseline, we evaluated three RL-only agents: a proprioceptive-tactile agent, a proprioceptive-only agent, and a proprioceptive-only variant which excludes the last action $\mathbf{a_{t-1}}$ (Figure 3). In *Find*, the proprioceptive-tactile agent offered only marginal sample efficiency gains, achieving equivalent final performance to the proprioceptive agent. The success of the proprioceptive agent critically relied on the last action, which enabled implicit object contact inference via control error, confirmed by policy inspection revealing a strategy to maximise collision probability. Conversely, in *Bounce*, adding tactile information yielded higher sample efficiency and superior returns, although the proprioceptive agent still attained high returns by developing a degenerate, state-agnostic bouncing motion with an outstretched hand. The *Baoding* task demonstrated the greatest utility for tactile information, driving the agent from complete failure to functional success, albeit with high variance. These task-dependent outcomes collectively suggest that tactile sensing only confers utility in specific scenarios, a theory we further develop in Section 6.

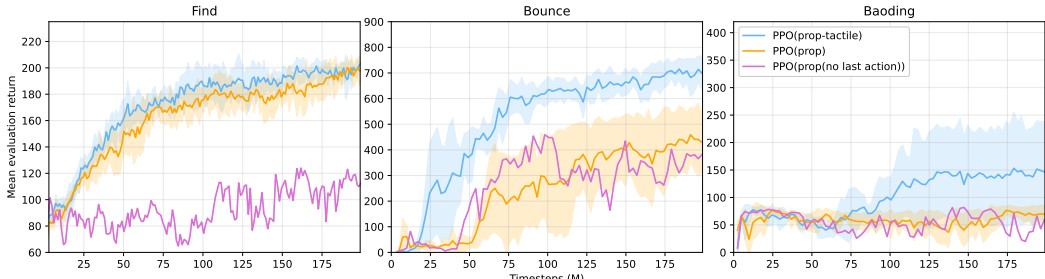

Figure 3: **RL-only.** Mean evaluation returns of proprioceptive-tactile ($\mathbf{o^{prop}} \oplus \mathbf{o^{tact}}$) and proprioceptive ($\mathbf{o^{prop}}$) agents. We run one seed with the last action $\mathbf{a_{t-1}}$ removed from the $\mathbf{o^{prop}}$ agent.

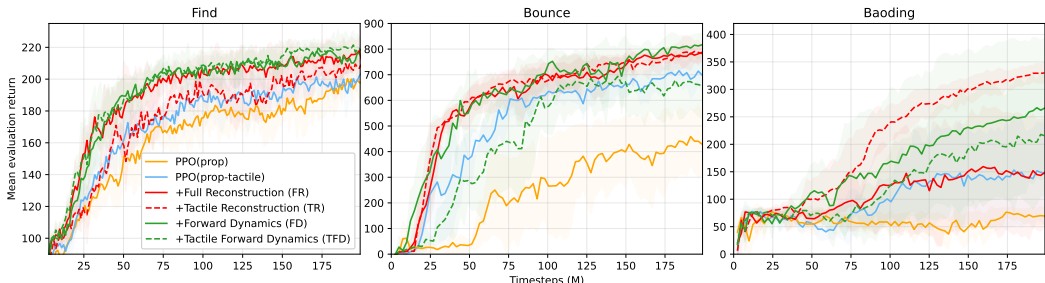

Figure 4: **RL+SSL.** Mean evaluation returns of the self-supervised agents.

**RL+SSL.** We evaluate our four proposed SSL objectives: Tactile Reconstruction (TR), Full Reconstruction (FR), Forward Dynamics (FD), and Tactile Forward Dynamics (TFD) (Section 3.2). As evident in Figure 4, agents trained with TR and FD consistently outperform RL-only baselines across all environments. Between these two leading methods, the FD agent yielded higher mean returns in *Find* and *Bounce*. Conversely, in *Baoding*, the TR agent achieved a higher mean return due to a tighter performance distribution, despite the FD agent reaching a higher performance upper bound. The performance of the remaining methods (FR and TFD) was sensitive to the specific environment, showing no consistent trend. For instance, TFD was the best objective in *Find* and the worst objective in *Bounce*. Further analysis is provided in Appendices A,D, and H.

**RL+SSL+Memory.** We investigated the impact of a separated auxiliary memory by applying it to the FD agent, selected for its superior general performance. To account for the widening data distribution, we conducted a targeted re-sweep over just the SSL hyperparameters and $N_{rollouts} \in 2, 3, 4$ (limited due to computational resources). As illustrated in Figure 5, the addition of the separated memory had only a minimal effect on returns in *Find* and *Bounce*, but led to a substantial improvement in *Baoding*. We hypothesise that this result stems from the inherent complexity of the *Baoding* task, which requires the agent to observe and act upon a longer temporal horizon of dynamics. A two-ball rotation inherently spans more timesteps than a bounce or contact event, potentially making the longer memory sequences more beneficial for learning.

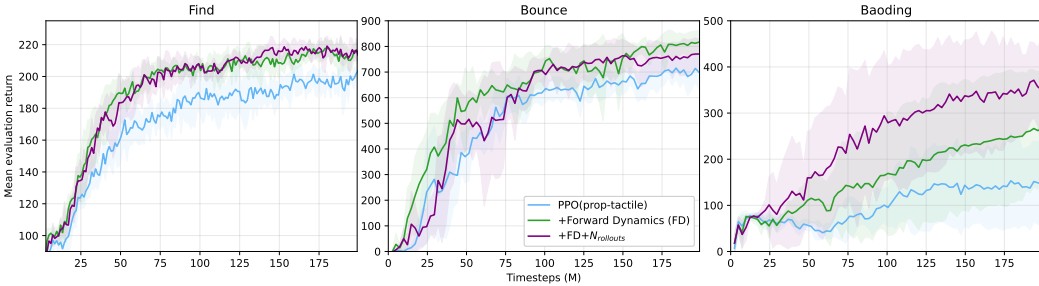

Figure 5: **RL+SSL+Memory.** Mean evaluation returns of the FD agent trained on the on-policy rollout vs $N_{rollouts}$ last rollouts. The effect is minimal in *Find* and *Bounce* but substantial in *Baoding*.

# 6 Discussion

**Q1: Do binary contacts offer benefits beyond proprioceptive history?**

Our results indicate that that binary contacts *do* offer benefits beyond proprioceptive histories, but the degree of utility is task-dependent. We hypothesise that explicit tactile information remains critically useful, but only in scenarios where proprioceptive joint control errors fail to reliably register all environment dynamics. For instance:

1. *Decoupled object-robot dynamics*: When an object's motion has a component orthogonal to the articulation direction, resulting in contact with negligible change in joint control error. For instance, in the *Baoding* task, the balls primarily move horizontally around the hand's plane, decoupled from the joint motion used for manipulation.

2. *Low-inertia objects*: Very light, deformable objects (e.g., the 30g ball in *Bounce*, a sheet of paper, or a sponge) do not exert a sufficiently large reactive force upon contact. Consequently, the robot's proprioceptive sensors fail to register a significant signal, necessitating reliance on explicit tactile sensing for reliable detection.

3. *Contact spatial ambiguity:* When the policy requires knowing the specific location of the contact along a single rigid link. The joint control error only provides the net force contribution to the motor, making it difficult to ascertain the distance of the force from the joint.

4. *Multi-contact resolution:* When the policy needs to differentiate the source of the total force contribution (i.e., whether it resulted from one strong contact or multiple simultaneous, weaker contacts). The control error only provides the total magnitude of the net force, blurring the distinction between multiple, fine-grained contact events.

**Q2: Why exactly is the self-supervision helping?**

We hypothesise that self-supervised learning helps by enforcing the compression of task-critical information into the learned representation $\mathbf{z_t}$. We test this hypothesis by measuring the mutual information $I(\mathbf{z_t}; \mathbf{s_t})$ between the reduced latent representation $\mathbf{z_t}$ and a vector of ground-truth state variables, $\mathbf{s_t}$. We utilise the Kraskov-Stoegbauer-Grassberger (KSG) estimator [27], following the approach for continuous variables as outlined in [15]. For all agents, we collect 5,000 samples of $(\mathbf{z_t}, \mathbf{s_t})$ pairs. The raw $\mathbf{z_t}$ is a 256-dimensional vector, which we reduce to $D = 13$ components via Principal Component Analysis (PCA) to counteract the bias of the KSG estimator in high dimensions. We use $K = 4$ nearest neighbors for estimation. For *Bounce*, $\mathbf{s_t}$ is the ball position, quaternion, linear velocity, angular velocity, and number of timesteps without contact. For *Baoding*, $\mathbf{s_t}$ is the balls' position, linear velocity, and relative distance. See Figure 6 for results.

In *Bounce*, the base PPO agent achieves the highest $I(\mathbf{z_t}; \mathbf{s_t})$, primarily because its successful policy results in a highly repetitive, low-entropy motion (trapping the ball), which artificially inflates the MI. The reconstruction agents had zero MI, while both dynamics agents registered non-zero MI. In *Baoding*, the distribution of MI across agents more closely mirrors the performance spread, with the FD agent capturing nearly triple the MI of the PPO agent. We also perform marginal MI analysis on individual state elements $I(\mathbf{z_t}; s_{t,j})$ to understand the specific encoding strategy used by each objective. For the *Baoding* environment, the FD agent was the only model to achieve non-zero marginals, successfully encoding the ball positions in order of priority: $x$ (parallel to the hand), $y$ (perpendicular to the hand), and $z$. In the *Bounce* environment, all agents recovered the number of timesteps without contact. While the tactile agents (TR and TFD) further recovered the ball vertical velocity, the FD agent proved the most effective, uniquely encoding both the ball vertical velocity and positional information ($x$ and $z$ coordinates).

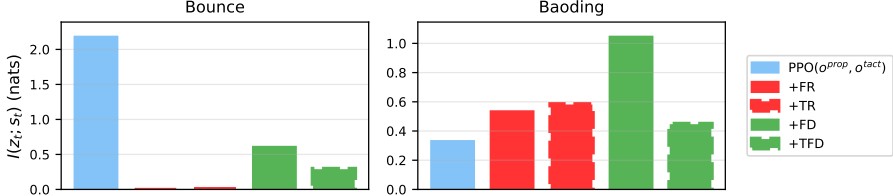

Figure 6: **Mutual information estimation**. We estimate $I(\mathbf{z_t}; \mathbf{s_t})$ between the reduced observation representation $\mathbf{z_t}$ and the ground-truth state $\mathbf{s_t}$ to evaluate the representation quality.

**Q3: Is self-supervision strengthening the proprioceptive, tactile, or combined representation?**

The influence of self-supervision on sensory representations differs significantly between the reconstruction and dynamics objectives, with its effect being highly dependent on the environment.

**Reconstruction.** The relative performance between FR and TR is different in all environments (Figure 4). In *Find*, FR outperforms TR, suggesting more benefit from compressing proprioceptive history since the agent was more reliant on the control errors for object detection. Conversely, in *Bounce*, FR and TR performed almost identically, implying the primary gain came from compressing the tactile signal. Most compellingly, in *Baoding*, TR far outperformed FR, demonstrating that combining proprioceptive reconstruction with tactile reconstruction resulted in negative interference. Given that only TR achieved zero failure runs, this suggests that agent failures are directly linked to an inability to robustly represent critical tactile information when its objective is combined with proprioception.

**Dynamics.** The effect of the dynamics prediction is less conclusive, as the comparison is between learning forward dynamics on the combined representation (FD), summed with a tactile reconstruction loss on the predicted latent state (TFD). TFD showed a minor benefit in *Find*, hinting at a useful combination of losses. However, in the more dynamic tasks of *Bounce* and *Baoding*, TFD performed worse than FD. This result suggests that explicitly forcing a separate tactile reconstruction loss is either redundant or counterproductive. Two primary explanations exist: 1) the FD objective on the combined representation is robust enough to implicitly capture the necessary tactile information within its learned state representation; or 2) adding an explicit reconstruction loss introduces an undesirable training conflict that compromises the overall predictive quality. Isolating the precise mechanism by which the tactile signal influences dynamics prediction will require further investigation, ideally utilising a distinct tactile-only forward model.

**Q4: How do the results correspond with physical metrics?**

The performance using physical metrics is shown in Figure 7, with the bold bars depicting the mean across seeds, and the shaded bars depicting the maximum across seeds. Overall, the performance improvements do correspond with meaningful changes in physical behaviours. For example, our best self-supervised agents on average find an object 36% faster (FD, 1.4 vs 1.9 seconds), bounce a ball 10 more times in 10 seconds (FD, 79 vs 69), and complete 17 Baoding rotations compared to 5 in 10 seconds (FD with auxiliary memory). We note that the best *Bounce* agent achieves 88 bounces in 10 seconds, beating the (human) Guinness World Record of $\sim 59$. Similarly, the best RL-only *Baoding* agent achieves 13 rotations in 10 seconds, on par with skilled humans, while the best self-supervised agent achieves 25. We fully acknowledge these results are not likely to transfer directly as-is into the real-world, but still include as a point of interest. See Appendix C for more dicussion of human performance.

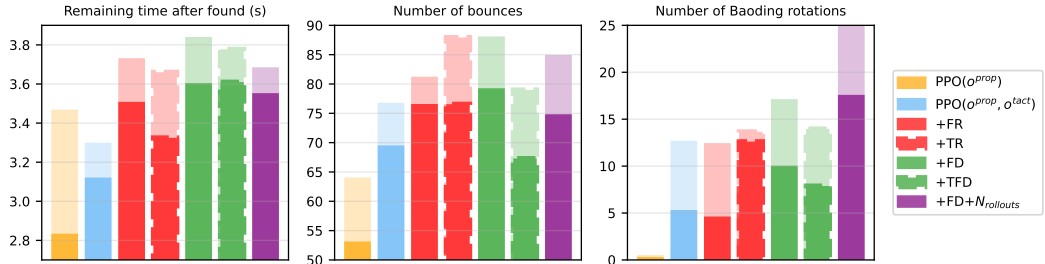

Figure 7: **Physical metrics.** Maximum (shaded) and mean (bold) across seeds.

**Q5: How well can a forward model learn the dynamics of tactile interactions?**

Very well. See Appendix D for plots of true positive rate, false negative rate, precision, and recall for up to 10 timesteps into the future for *Bounce* and *Baoding*. There we also also provide a spatio-temporal visualisation of predicted vs actual next tactile states (Figures A6 & A7). From analysing these figures, we observe there is a slight tendency to overpredict rather than underpredict contacts, likely due to the high positive weighting we apply in the BCE loss ($p_c = 10$). Despite having access to a history of tactile states in the current observation, some of these states are not always perfectly "copied" over. Interestingly, from the no-contact observation $o_7$ in *Bounce*, the decoder correctly

anticipates contact in the next state (albeit in the wrong locations). This result aligns with the findings in **Q2**, where we show the representation is encoding the ball $z$ position and velocity component, which is sufficient to predict future contacts from. Also in *Baoding*, the some of the false positive predictions in states $\mathbf{o_{10}}, \mathbf{o_{11}}$ are just one-step too early.

**Q6: Can on-policy agents benefit from learning representations on off-policy data?**

The results in Figure 5 provide evidence for *yes*, with the effect ranging from minimal (*Find*, *Bounce*) to substantial (*Baoding*). We think the impact was pronounced on *Baoding* because of the rotation task requiring reasoning over a longer temporal horizon. These findings highlight a promising research direction for incorporating off-policy data into on-policy training pipelines, potentially bridging the gap between the strengths of both paradigms.

**Q7: How do our findings translate to practical recommendations?**

We compress our findings into two recommendations. Our work has demonstrated that "blind" robotic agents trained jointly with self-supervision outperform RL-only agents across a diverse range of control tasks. Thus, our first recommendation is to train tactile-based RL agents jointly with tactile reconstruction or forward dynamics with a separated auxiliary memory, if you are working in a similar setting and would like to get a higher (and potentially more reliable) distribution of returns. Second, while we acknowledge that increased sensory information is theoretically advantageous, it incurs substantial computational costs as well as being statistically harder i.e., the space of functions over $Z^{N_{\text{sensors}}}$ is much larger than $\{0, 1\}^{N_{\text{sensors}}}$. In addition, the bandwidth of pixel-based signals directly limits the number of parallel environments that can be executed in Isaac Lab and other simulators. Since our work has revealed unexpected efficacy using binary tactile observations, we recommend initially implementing simpler tactile information formats (e.g., binary, continuous, or contact pose), and switching to pixel-based tactile representations only if needed.

## 7 Limitations

The primary limitation of this work is the absence of validation on physical robot hardware. However, we have minimised the sim-to-real gap by focusing on sparse binary contact signals, which avoids major complexities associated with continuous sensor noise. Crucially, our core contributions are optimisation strategies designed to yield superior and distinct policies. The benefits of these strategies should therefore be transferable, provided the final sim-to-real gap for the RL-only agent can be bridged. We also note that training self-supervised agents increases computation time compared to RL-only. The effect is less noticeable for reconstruction, but becomes more dramatic the higher the value of sequence length $n$ in forward dynamics. In addition, a limitation of using a separated auxiliary memory is that more memory is required. Regarding generalisation, we would expect to see similar results if our approach is applied to other environment domains (e.g., locomotion).

**Broader impacts.** Our work advances the capabilities of autonomous sensory-driven robotic agents in simulated environments. While real-world deployment down the line could positively impact healthcare and assisted living, we acknowledge potential negative consequences including workforce displacement and unintended applications, necessitating responsible development and appropriate regulatory oversight.

**Acknowledgments.** EM is supported by a Ph.D. studentship from the UK Engineering and Physical Sciences Research Council (EPSRC) via the Robotics and Autonomous Systems CDT. We thank Dr. Stefano V. Albrecht for providing formative guidance and initial conceptualisation of this research.

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

# Appendix

## A  Results as physical metrics

Figure A1 shows the mean number of seconds it takes the agent to locate the object within different tolerances. The distance $d$ is measured between the end-effector (imaginary fixed frame in the center of the parallel-jaw gripper) and the object center. While the performance between a proprioceptive and proprioceptive-tactile agent is similar for a $d = 3$ cm threshold, the benefits of tactile data become more pronounced with smaller tolerances. The relative performance between SSL objectives is consistent except for $d = 0.05$ cm, where FR and TFD are the fastest. Figure A2 shows the number of bounces the agent achieves in 10 seconds throughout learning, and Figure A3 shows the number of complete Baoding rotations achieved. Due to the overlapping performance distributions in *Baoding*, we provide alternative figure versions with only a subset of runs and/or no bad seeds. Across all seeds, from Figure A3 (bottom left) we can see applying tactile reconstruction or full dynamics self-supervision approximately doubles the number of complete rotations achieved.

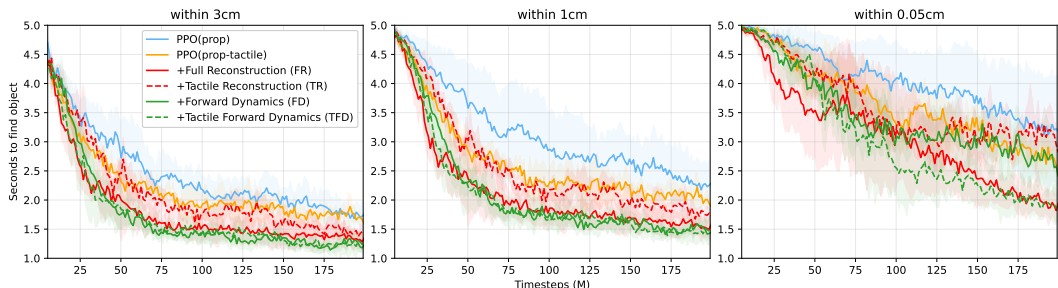

Figure A1: **Number of seconds to locate the object in *Find*.** We show results for within 3, 1, and 0.05 cm distance tolerances.

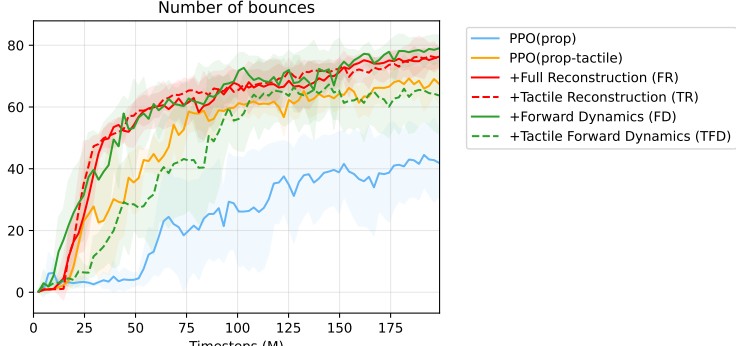

Figure A2: **Number of bounces in 10 seconds in *Bounce*.** A bounce is defined as a contact after 5 timesteps of no contact.

## B  Impact of auxiliary memory on self-supervised learning

Figure A4 shows the average minibatch (full) dynamics loss of a *Baoding* agent with varying auxiliary memory size. There are 19 rollouts between each peak, and with a rollout length of 32 steps this corresponds to 608 timesteps $\sim 1$ episode. We interpret the loss peaks coming from episode resets when the Baoding balls are set above the palm and potentially spend a large portion of one rollout falling into the palm, disrupting SSL as the loss of tactile data is highly out-of-distribution. Increasing the memory size flattens and widens out the peak, potentially leading to smoother gradient updates and the improved agent performance in this task.

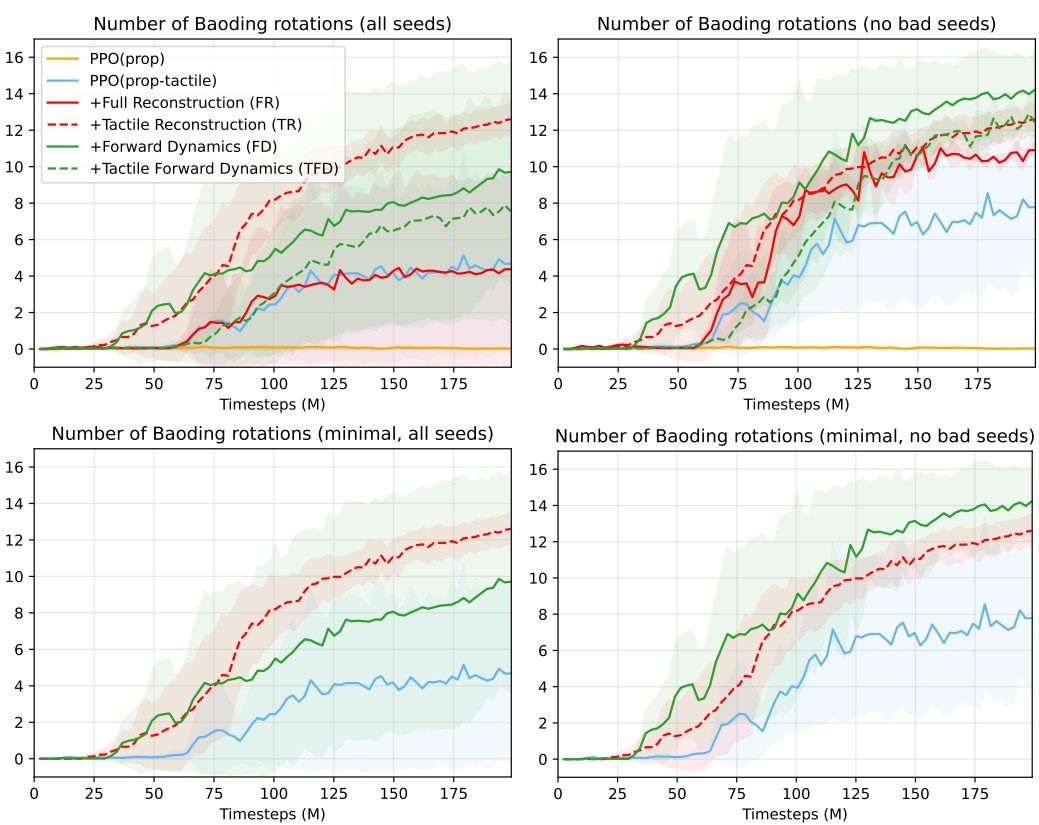

Figure A3: **Number of complete rotations in *Baoding* in 10 seconds**. A rotation is when each ball returns to its initial position. *Left:* The distribution across all seeds. *Right:* The distribution across seeds that learned at least 1 rotation. *Top:* All experiments. *Bottom:* A subset of experiments.

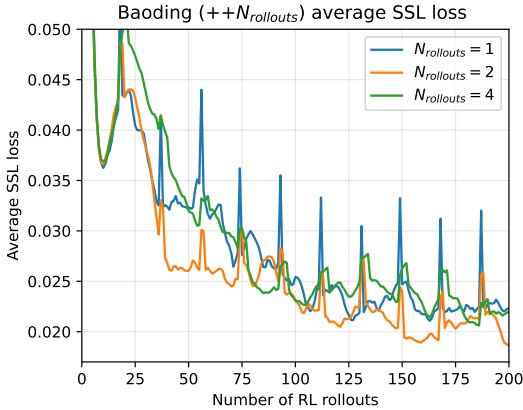

Figure A4: **Auxiliary loss vs memory size.** Increasing the auxiliary memory size reduced the spike amplitude of the mean minibatch self-supervised dynamics loss of a *Baoding* agent.

## C   Human capabilities

*Find.* Replicating the environment in real life and testing with one human subject, the average time to find and grasp the object across 10 trials was 2.1 seconds.

*Bounce.* The most tennis ball touches using the hand in one minute is 353, and was achieved by Manikandan Thirumaniselvam in India on 4 February 2023[1]. This translates to $353/6 \sim 59$ bounces in 10 seconds. We note the properties of our ball (30g, 70mm diameter) are different to a tennis ball ($\sim$ 58g, 67mm diameter), but would expect to see similar results.

*Baoding.* The fastest demonstration we could find online achieves 13 rotations in 10 seconds[2]. We believe the properties of our Baoding balls and the ones in the video are identical (we acquired the same ones and took measurements: 55g, 1.5 inch diameter).

## D   Future tactile state prediction analysis

To understand how well MLPs can model the forward dynamics of tactile interactions, we provide various classification metrics throughout training (Figure A5) and spatio-temporal rollout visualisations of trained *Bounce* and *Baoding* agents (Figures A6 and A7). We can compute metrics such as recall and precision since tactile reconstruction was formulated as classification (rather than regression), but unlike typical supervised learning the performance does not increase monotonically because of the nonstationary data distribution.

Overall, we were surprised how robust the performance remained $n - 1$ timesteps into the future (evidenced by how difficult it is to distinguish between the timesteps in Figure A5). This suggests that the multi-step dynamics objective was very effective at encoding information relevant for future state predictions. Moreover, the rate of missed contacts was $<1\%$ throughout training for both *Bounce* and *Baoding*, which we attribute to the high positive weighting applied. Interestingly, despite the agent having access to the 3 last tactile readings that would form the first 3 tactile readings of the prediction, some of these states were not always perfectly "copied" over (e.g., $\mathbf{o_5}$, $\mathbf{o_{11}}$, and $\mathbf{o_{12}}$ in *Bounce*). This highlights that the readings are not merely being 'memorised', but being represented in some (imperfect) way. For future work, we believe dynamically updating the positive weighting as the inverse mean of the minibatch would be beneficial to reflect the nonstationary training distribution. Additionally, our implementation applied the same weighting to each activation region, but some regions are much more active than others (e.g., compare palm to pinky in Figure A7), and this discrepancy should be accounted for.

---

[1]**Bounce record**: https://www.guinnessworldrecords.com/world-records/590513-most-tennis-ball-touches-using-the-hand-in-one-minute, **Bounce video**: https://www.youtube.com/watch?v=ORiHY0MwT4A

[2]**Baoding video**: https://www.youtube.com/shorts/x-ns-auc098

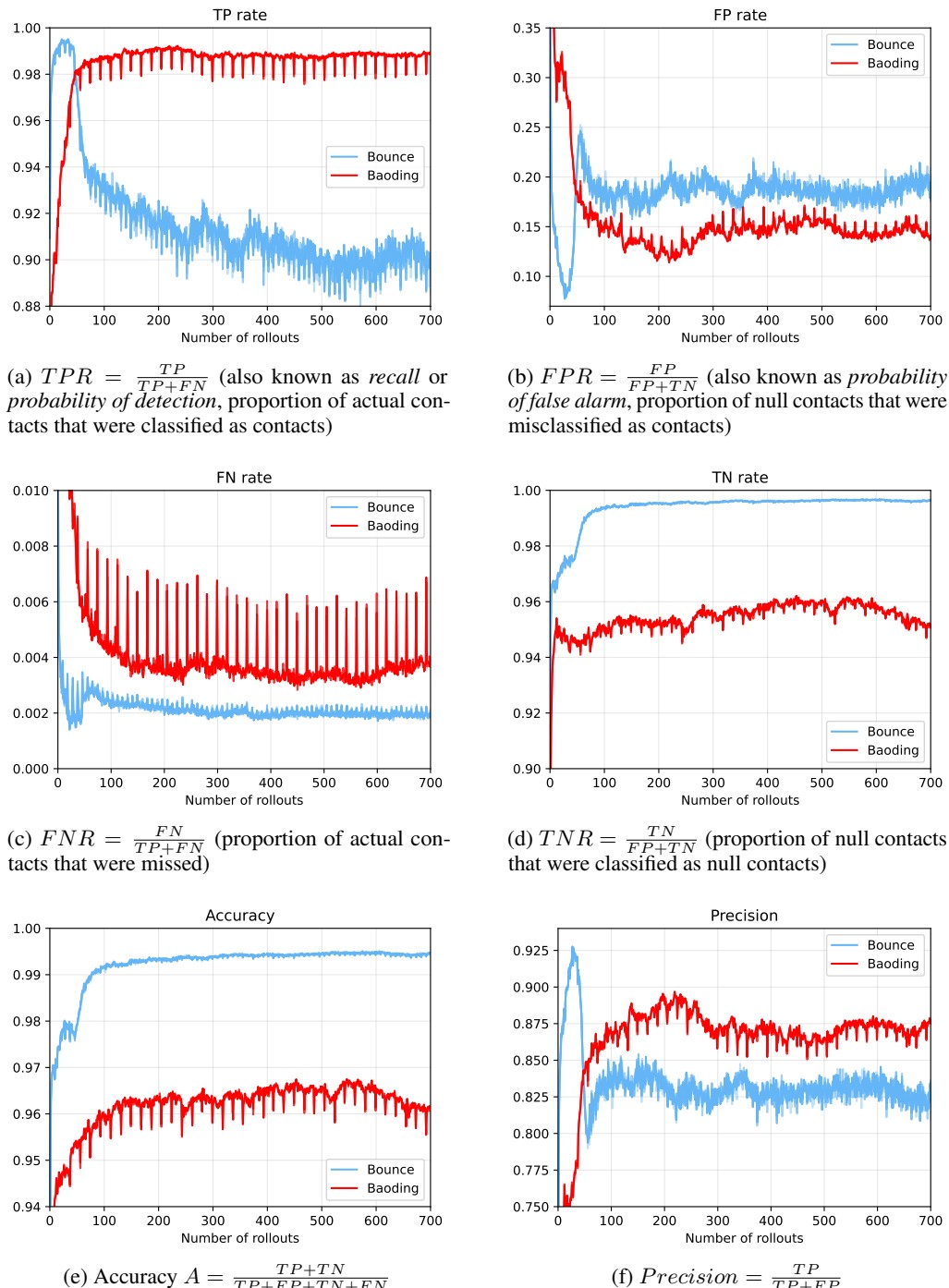

(a) $TPR = \frac{TP}{TP+FN}$ (also known as *recall* or *probability of detection*, proportion of actual contacts that were classified as contacts)

(b) $FPR = \frac{FP}{FP+TN}$ (also known as *probability of false alarm*, proportion of null contacts that were misclassified as contacts)

(c) $FNR = \frac{FN}{TP+FN}$ (proportion of actual contacts that were missed)

(d) $TNR = \frac{TN}{FP+TN}$ (proportion of null contacts that were classified as null contacts)

(e) Accuracy $A = \frac{TP+TN}{TP+FP+TN+FN}$

(f) $Precision = \frac{TP}{TP+FP}$

Figure A5: **Classification metrics for TFD agents.** We compare the predicted vs true future tactile states of *Bounce* and *Baoding* agents trained with TFD for $t = 1, 2, 3, 9$ and $t = 1, 2$ timesteps into the future respectively. The metrics for future timesteps are shown with decreasing opacity but strongly overlap with $t = 1$, making it difficult to distinguish. The metrics were computed on one minibatch after the learning update.

For *Bounce*, tactile interactions are increasingly sparse (e.g., compare Figures A6 and A7). Thus the metrics we were most interested in were true positive rate (proportion of contacts caught) and false negative rate (proportion of contacts missed). True positive rate (TPR) decreased from ∼99% to 90% throughout training, which we attribute to increased difficulty predicting the landing sites of a bouncing ball. The proportion of contacts that were missed (FNR) was surprisingly low throughout training, converging to ∼ 0.2%. Fascinatingly, from the no-contact observation $\mathbf{o_7}$, the decoder correctly anticipates contact in the next state (albeit in the wrong locations, however two predictions are just 1 timestep early). This result suggests that object height information is being encoded.

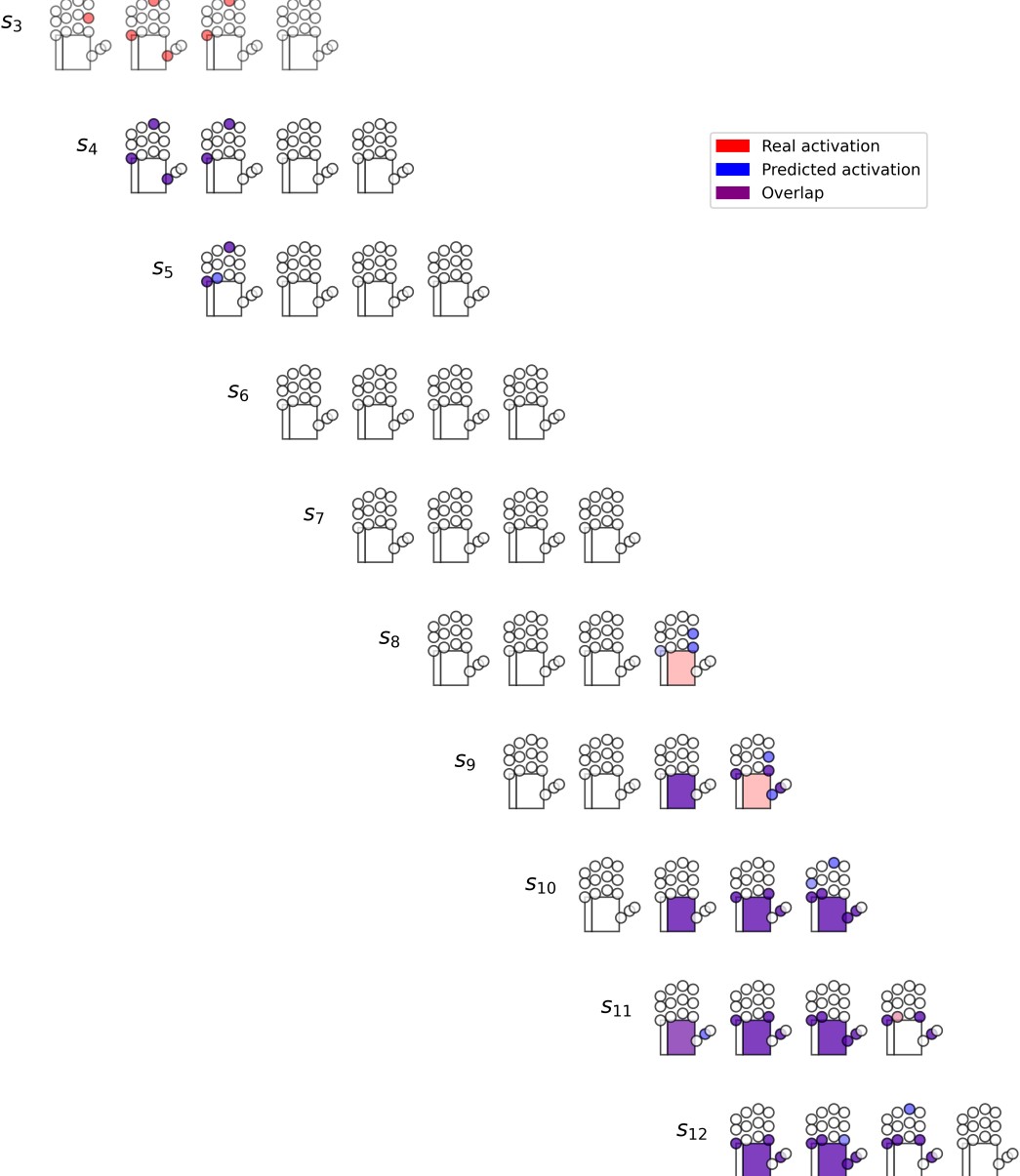

Figure A6: **Spatio-temporal visualisation of 1-step predicted future tactile states of a TFD** ***Bounce* agent**. The tactile observation $\mathbf{o_t^{tact}}$, displayed here as $s_t$, is comprised of $k = 4$ past readings, and each column corresponds to the same reading. The sigmoid activation of the tactile prediction is displayed with varying opacity (e.g., less confident is lighter blue).

For *Baoding*, the frequency of tactile interactions remains high throughout learning, thus all metrics are of relevance. The proportion of contacts that were correctly detected was ~99%, which is very high. Similarly to *Bounce*, very few true contacts were missed (FNR), and the false positive rate was relatively high at ~15%. The accuracy remained at >96% for majority of training. Also similarly to *Bounce*, some of the false positive predictions (e.g., in observations $o_{10}, o_{11}$) are just one-step too early. Finally, across all metrics in Figure A5, we can see negative performance spikes at fixed intervals throughout learning that are not present for *Bounce*. This is discussed in Appendix B.

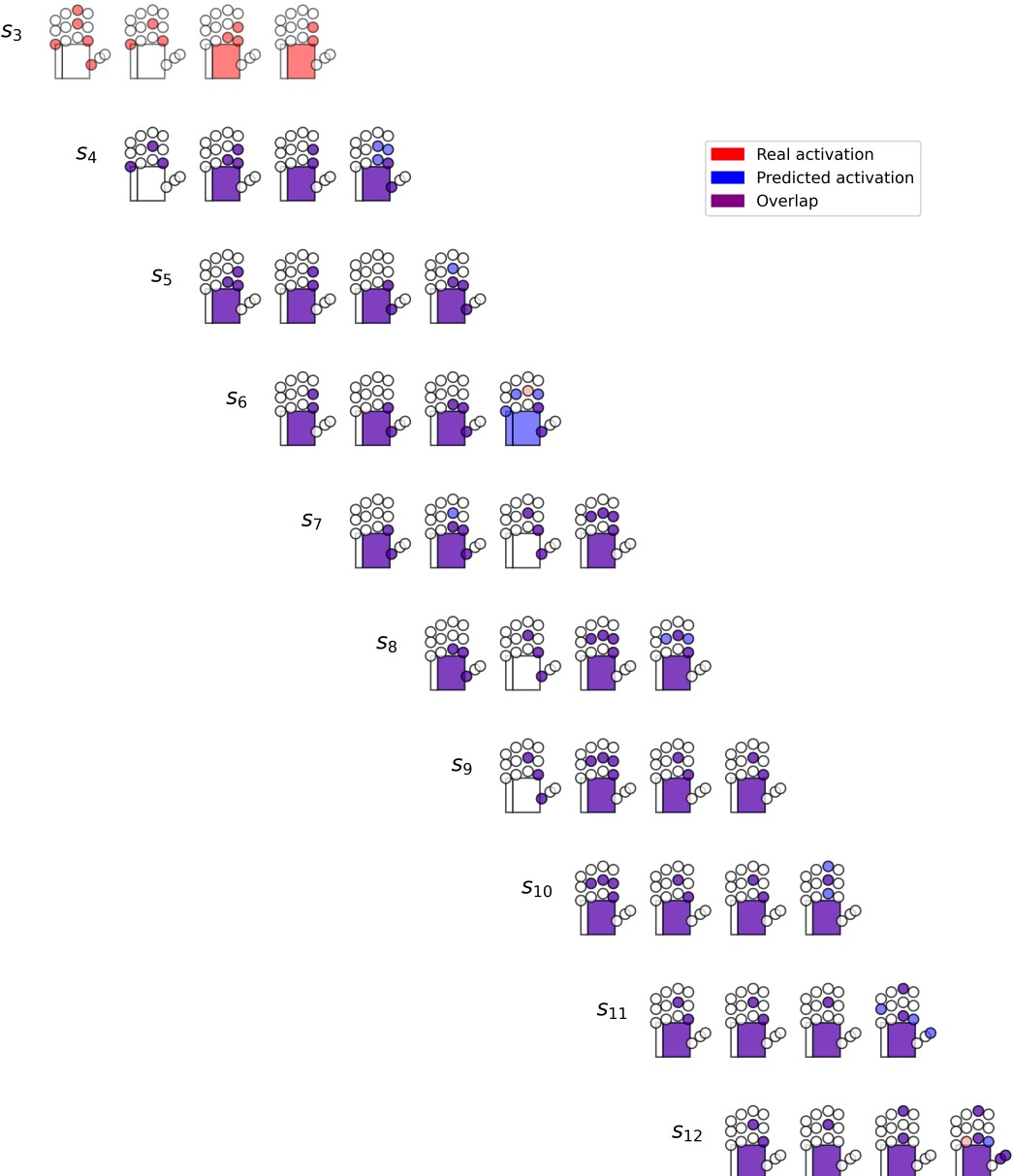

Figure A7: **Spatio-temporal visualisation of 1-step predicted future tactile states of a TFD** ***Baoding* agent**. The tactile observation $o_t^{tact}$, displayed here as $s_t$, is comprised of $k = 4$ past readings, and each column corresponds to the same reading. The sigmoid activation of the tactile prediction is displayed with varying opacity (e.g., less confident is lighter blue).

# E   POMDP

## E.1   Observations

A summary of the observations is shown in Table A1. Each environment uses a $k$-length history of sensor readings to form the observation $\mathbf{o_t}$ ($k = 16$ for *Find* and $k = 4$ for *Bounce* and *Baoding*, found empirically). We apply input preprocessing as follows: joint angles $\boldsymbol{\theta}$ are normalised between [-1,1]. Joint velocities $\dot{\boldsymbol{\theta}}$ are scaled down by a scalar (0.33 for Franka, 0.2 for Shadow). We retrieve the net contact forces through Isaac Lab's `ContactSensor` class[3] and binarise it. To mimic real-world tactile sensing and make the task more challenging, we registered two 'plate-like' bodies to atop the Franka fingers to act as contact sensors (Figure A8). With this setup, the sensors would only register forces that resulted from collision with these bodies, which is only possible from the object or other finger, and not the ground. Since the Shadow Hand was fixed in midair we let each link become a sensor, resulting in 17 sensors (Figure 1).

Table A1: Observation spaces across environments.

| Type | Symbol | Description | *Find* | *Bounce* | *Baoding* |
|---|---|---|---|---|---|
| $\mathbf{o^{tact}}$ | $\mathbf{b}$ | binary contacts | 2 | 17 | 17 |
| $\mathbf{o^{prop}}$ | $\mathbf{a_{t-1}}$ | last action | 9 | 20 | 20 |
| | $\boldsymbol{\theta}$ | joint angles | 9 | 24 | 24 |
| | $\dot{\boldsymbol{\theta}}$ | joint velocities | 9 | 24 | 24 |
| | $\mathbf{x_{EE}}$ | EE position | 3 | | |
| | $\mathbf{q_{EE}}$ | EE quaternion | 4 | | |
| | $w$ | gripper width | 1 | | |
| | $\mathbf{o_t}$ | observation | $37{\times}k$ | $85{\times}k$ | $85{\times}k$ |

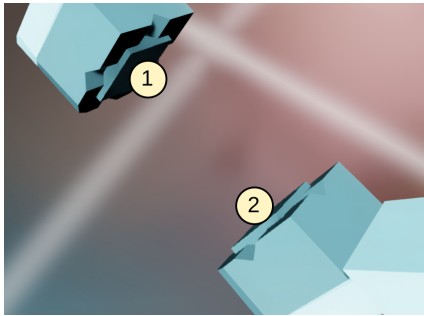

Figure A8: **Contact sensors for *Find*.** We placed two bodies on Franka's fingers to act as contact sensors.

## E.2   Actions

All robots are joint position controlled, with $\mathbf{a_t} \in \mathbb{R}^9$ for Franka and $\mathbf{a_t} \in \mathbb{R}^{20}$ for the Shadow Hand. The Shadow Hand is underactuacted, with coupled distal and proximal joints (2 wrist + 5 thumb + 3 index + 3 middle + 3 ring + 3 pinky + 1 metacarpal = 20).

## E.3   Rewards

The rewards for each environment step are given by the sum of the different terms each multiplied by the given scale. For enhanced value function learning, we track a running mean and variance for normalising the returns and values in the PPO update.

For *Find* there is one reward term $r_{dist}$ that grows as distance between the object and end-effector decreases.

---

[3]https://isaac-sim.github.io/IsaacLab/main/source/overview/core-concepts/sensors/contact_sensor.html

For *Bounce*, the reward is given by $r_{air} + r_{bounce} + r_{fall}$. To aid initial exploration, the agent is rewarded proportionally to number of time steps since last contact, $r_{air}$. A bounce event is defined as there being contact after 5 timesteps of no contact, for which there is a bonus $r_{bounce}$. The fall penalty is applied if the object is more than 24 cm from a fixed center position.

For *Baoding* the reward is given by $r_{dist_1} + r_{dist_2} + r_{rotation} + r_{fall}$. Our original approach was to maximise the $xy$ angular velocity of the vector connecting the two balls which worked well, but sometimes the agent came up with creative strategies that no longer resembled the original Baoding task. Thus, we reformulated the reward around two fixed target poses (Figure A9). When the centers of both balls were within 1.0 cm of the given target centers, the targets switched and the agent receives a bonus reward $r_{rotation}$. We also used two dense ball-center-to-target distance rewards $r_{dist_1}, r_{dist_2}$ to aid exploration in the beginning. This approach better constrained the ball positions and stabilised policies for comparing methodologies. The fall penalty is applied if the distance between the balls exceeds 15 cm.

Table A2: Reward components across environments.

| Symbol | Description | Equation | Scale | *Find* | *Bounce* | *Baoding* |
|--------|-------------|----------|-------|--------|----------|-----------|
| $r_{dist}$ | distance to target | $1 - \tanh d_{target}/0.1$ | 1, 0.1 | ✓ | | ✓✓ |
| $r_{air}$ | time without contact | $+1$ | 0.01 | | ✓ | |
| $r_{bounce}$ | successful bounce | $+1$ | 10 | | ✓ | |
| $r_{rotation}$ | successful rotation | $+1$ if $d_{1\&2} < 1.0$cm | 10 | | | ✓ |
| $r_{fall}$ | fall penalty | $-1$ if $d > d_{max}$ | 10 | | ✓ | ✓ |

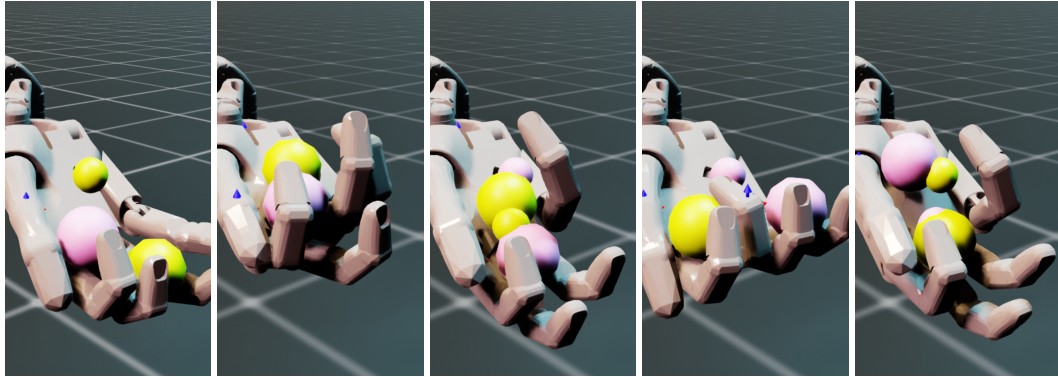

Figure A9: **Virtual targets in *Baoding***. When both balls are within 1cm of their virtual targets (shown as smaller balls), the targets switch and the agent recieves a bonus.

### E.4 Reset

Episodes can be terminated by a failure state, or truncated by a time limit. The *Find* environment episode length is $T = 300$ timesteps (5s), and *Bounce* and *Baoding* environments are $T = 600$ timesteps (10s). The *Bounce* and *Baoding* episodes can be terminated early if the balls fall out of the hand, measured by a distance. At the beginning of each episode, the Franka joint angles are randomised up to $\pm7°$ and the Shadow joint angles are randomised upto $\pm 20\%$. The ball in *Find* is randomised to any position on the 20cm $\times$ 20cm plate. The *Bounce* ball and *Baoding* balls are randomised by $\pm1$ cm and $\pm0.5$cm respectively along the global $xyz$ axis .

## F   Network architectures

**Encoder.** The encoder $\mathbf{e}$ is a 3-layer MLP with dimensions $\mathbf{o_t} \to 1024 \to 512 \to 256 \to \mathbf{z_t}$. Layer normalisation and ELU activations are applied after each layer.

**Policy.** The policy $\pi$ is a 3-layer MLP with dimensions $\mathbf{z_t} \to 128 \to 64 \to n_{actions} \to \mathbf{a_t}$. ELU activations are applied after the first two layers. The output layer activation is $\tanh$.

**Value function.** The value function $\mathbf{v}$ is a 3-layer MLP with dimensions $\mathbf{z_t} \rightarrow 128 \rightarrow 64 \rightarrow 1 \rightarrow V(\mathbf{o_t})$. ELU activations are applied after the first two layers. The output layer activation is identity.

**Reconstruction.** The decoder $\mathbf{d}$ is a 3-layer MLP. The full reconstruction decoder has dimensions $\mathbf{z_t} \rightarrow 512 \rightarrow 512 \rightarrow \texttt{len}(\mathbf{o_t}) \rightarrow \hat{\mathbf{o}}_\mathbf{t}$. The tactile reconstruction decoder has dimensions $\mathbf{z_t} \rightarrow 512 \rightarrow 512 \rightarrow \texttt{len}(\mathbf{o_t^{tact}}) \rightarrow \hat{\mathbf{o}}_\mathbf{t}^\mathbf{tact}$. ELU activations are applied after the first two layers. The output layer activation is sigmoid for tactile predictions, and identity for proprioception predictions.

**Forward dynamics.** The forward model $\mathbf{f}$ is a 3-layer MLP with dimensions $(\mathbf{z_t}, \mathbf{a_t}) \rightarrow 512 \rightarrow 256 \rightarrow 256 \rightarrow \hat{\mathbf{z}}_\mathbf{t+1}$, with ELU activations after the first two layers. The projector $\mathbf{p}$ is a 2-layer MLP with dimensions $\hat{\mathbf{z}}_\mathbf{t+i} \rightarrow 256 \rightarrow 256 \rightarrow \mathcal{L}_i$ with an ELU activation after the first layer. The target encoder $\mathbf{e_T}$ is identical to $\mathbf{e}$, but updated according to Equation 7 with $\tau = 0.01$.

$$\theta_{e_T} \leftarrow (1 - \tau)\theta_{e_T} + \tau\theta_e \tag{7}$$

# G    Hyperparameter tuning

We carefully tuned all our experiments to give each agent the best shot. For fairness, we followed the same hyperparameter tuning recipe for each individual experiment (3 environments × 7 experiments = 21 sweeps). We use the Optuna library [3] with the TPE sampler (5 startup trials) and no pruner. We wait for each sweep to reach 20 complete trials (some hyperparameter combinations lead to policy/value NaNs which are terminated early). The hyperparameters and possible ranges we tested are provided in Table A3, with the optimised values in Table A4. We did not sweep over the following hyperparameters: discount factor $\gamma = 0.99$, value loss scale $c_v = 0.1$, gradient norm clip 1.0, value clip 0.2, ratio clip 0.2. We provide the mean evaluation returns of each run in the sweep of *Bounce* with full dynamics self-supervision in Figure A10 to demonstrate the sensitivity of our agents to hyperparameters, and illustrate the importance of performing quality hyperparameter tuning for each individual experiment.

Table A3: Tunable hyperparameters and ranges for each experiment.

| Hyperparameter | Symbol | Tunable values |
|---|---|---|
| Rollout | $R$ | $\{16, 32, 64\}$ |
| Minibatches | $mb$ | $\{4, 8, 16, 32, 64\}$ |
| Learning epochs | $le$ | $\{4, 8, 16, 32\}$ |
| Learning rate | $lr$ | $[10^{-5}, 10^{-3}] \subset \mathbb{R}$ |
| Entropy loss scale | $c_{ent}$ | $\{0, 0.05, 0.1\}$ |
| Auxiliary learning rate | $lr_{aux}$ | $[10^{-5}, 10^{-3}] \subset \mathbb{R}$ |
| Auxiliary loss weight | $c_{aux}$ | $[10^{-3}, 10] \subset \mathbb{R}$ |
| Dynamics sequence length | $n$ | $\{2, 3, 4, 10\}$ |
| Auxiliary memory size | $N_{rollouts}$ | $\{2, 3, 4\}$ |

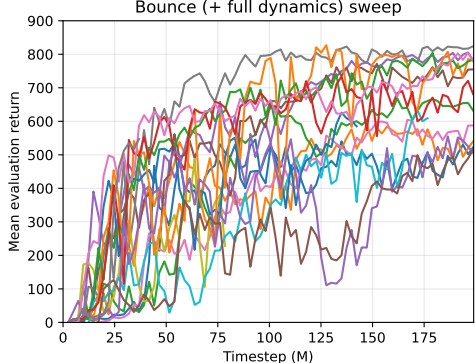

Figure A10: **Example *Bounce* sweep.** All experiments were highly sensitive to different hyperparameter combinations and seeds.

Table A4: Tuned hyperparameters for each experiment.

| Environment | Experiment | $R$ | $mb$ | $le$ | $lr$ | $c_{ent}$ | $lr_{aux}$ | $c_{aux}$ | $n-1$ | $N_{rollouts}$ |
|---|---|---|---|---|---|---|---|---|---|---|
| *Find* | PPO(prop) | 32 | 16 | 8 | $1.06 \times 10^{-5}$ | 0 | | | | |
| | PPO(prop-tactile) | 32 | 16 | 8 | $1.06 \times 10^{-5}$ | 0 | | | | |
| | +FR | 64 | 64 | 4 | $7.39 \times 10^{-5}$ | 0 | $5.91 \times 10^{-5}$ | 0.0023 | | |
| | +TR | 64 | 16 | 8 | $1.36 \times 10^{-5}$ | 0.1 | $2.55 \times 10^{-5}$ | 0.004477 | | |
| | +FD | 64 | 64 | 4 | $1.15 \times 10^{-5}$ | 0.1 | $1.55 \times 10^{-4}$ | 0.0062 | 2 | |
| | +TFD | 64 | 64 | 4 | $2.32 \times 10^{-5}$ | 0 | $1.57 \times 10^{-4}$ | 0.0024563 | 4 | |
| | +FD+$N_{rollouts}$ | 64 | 64 | 4 | $1.15 \times 10^{-5}$ | 0.1 | $3.81 \times 10^{-5}$ | 0.1364 | 3 | 3 |
| *Bounce* | PPO(prop) | 32 | 32 | 4 | $5.93 \times 10^{-5}$ | 0 | | | | |
| | PPO(prop-tactile) | 16 | 8 | 4 | $3.21 \times 10^{-4}$ | 0 | | | | |
| | +FR | 64 | 16 | 16 | $1.88 \times 10^{-4}$ | 0 | $2.77 \times 10^{-5}$ | 0.05669 | | |
| | +TR | 64 | 32 | 16 | $4.65 \times 10^{-5}$ | 0 | $5.13 \times 10^{-5}$ | 0.00384 | | |
| | +FD | 32 | 64 | 4 | $1.50 \times 10^{-4}$ | 0 | $4.53 \times 10^{-5}$ | 0.8462 | 10 | |
| | +TFD | 64 | 32 | 8 | $1.22 \times 10^{-4}$ | 0.05 | $1.64 \times 10^{-4}$ | 0.23547 | 10 | |
| | +FD+$N_{rollouts}$ | 32 | 64 | 4 | $1.50 \times 10^{-4}$ | 0 | $1.16 \times 10^{-4}$ | 0.19954 | 4 | 2 |
| *Baoding* | PPO(prop) | 32 | 8 | 4 | $9.96 \times 10^{-5}$ | 0.05 | | | | |
| | PPO(prop-tactile) | 32 | 4 | 8 | $2.02 \times 10^{-4}$ | 0.05 | | | | |
| | +FR | 16 | 32 | 4 | $3.68 \times 10^{-4}$ | 0 | $5.18 \times 10^{-5}$ | 0.058866 | | |
| | +TR | 64 | 32 | 8 | $3.61 \times 10^{-4}$ | 0 | $1.00 \times 10^{-5}$ | 0.2707 | | |
| | +FD | 32 | 16 | 4 | $5.47 \times 10^{-4}$ | 0 | $2.87 \times 10^{-4}$ | 3.686 | 2 | |
| | +TFD | 32 | 16 | 8 | $2.08 \times 10^{-5}$ | 0.05 | $1.53 \times 10^{-4}$ | 0.04839 | 3 | |
| | +FD+$N_{rollouts}$ | 32 | 16 | 4 | $5.47 \times 10^{-4}$ | 0 | $1.67 \times 10^{-5}$ | 1.6349 | 4 | 4 |

# H   Latent trajectory analysis

Figures A11, A12, and A13 show a two dimensional latent representation of a single episode across all environments. Trajectories for RL-only and a subset of self-supervised agents are shown (TR, FD, and TFD). The 256-dim observation representation $\mathbf{z_t}$ at each timestep was reduced using 2-component Principal Component Analysis (PCA). Note that the tactile activations shown are only the sum of activations in the current reading $\mathbf{o^{tact^t}}$, and does not sum across the reading history.

**Baoding**. The ring-like trajectory of the RL-only agent illustrates the repeated motion the agent develops. There are two tactile peaks on opposite sides of the ring, indicating symmetry in contact activations between half-rotations. The trajectory of the TR agent is quite different (heart-shaped, diffuse). This shows each rotation is slightly different, and there is now asymmetry between the contact activations of half-rotations. From rendering the policy, the gait is smooth like the FD agent but keeps the balls close together like the RL-only agent. The trajectory of the FD agent is again ring-like, but with tighter bounds than the RL-only agent. Like the TR agent, there is contact activation asymmetry between half-rotations. Finally, the TFD agent trajectory appears to be a blend of the FD and TR trajectories.

**Bounce**. The latent trajectories of the self-supervised agents are highly different to the RL-only agent. From the trajectory with the time colourbar, we can see that the sequential latent states of the RL-only agent are highly discontinuous and far apart (e.g., yellow), and the agent repeats the same motion with high precision. There are two regions with non-zero tactile observations of upto 6 activations, which is understood by the 'safe' gait of raising the index and pinky finger to stabilise the ball. The gait changes completely for the self-supervised agents, which predominately use 1 or 2 contacts. Sequential latent states are still spread out in various regions, but these regions are much more diffuse than in the RL-only.

**Find**. It is clear the self-supervised agents find the object faster by observing the trajectories colourised by time. Otherwise, the shape of the latent trajectories is not drastically different between RL-only and self-supervised agents. A distinction between 1 and 2 tactile activations appears in the FD agent trajectory.

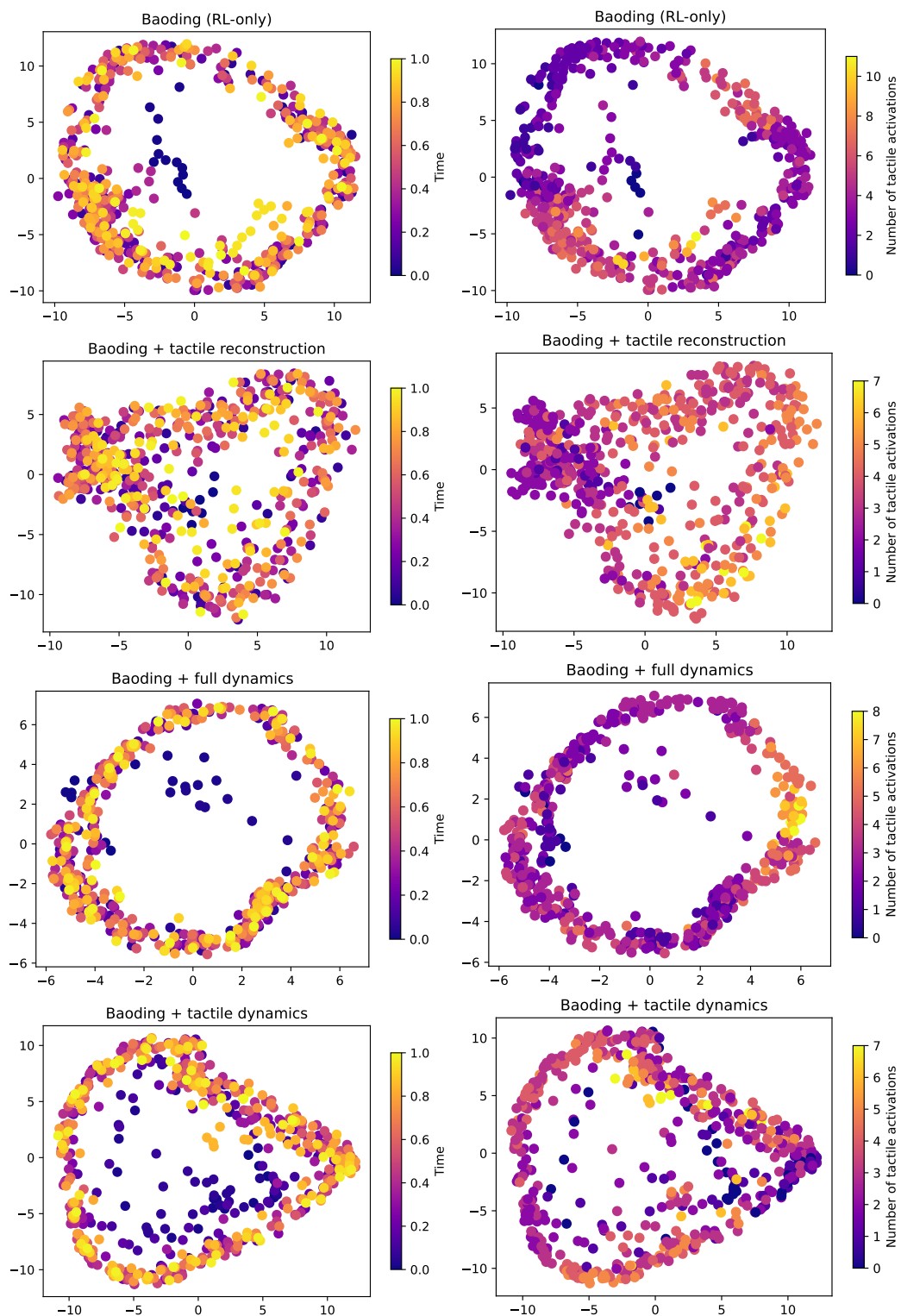

Figure A11: **Latent episode trajectory (PCA) of RL-only, TR, FD, and TFD *Baoding* agents.** *Left:* Samples colourised by time. *Right:* Samples colourised by summed binary contacts of the last tactile reading $\mathbf{o^{tact^t}}$.

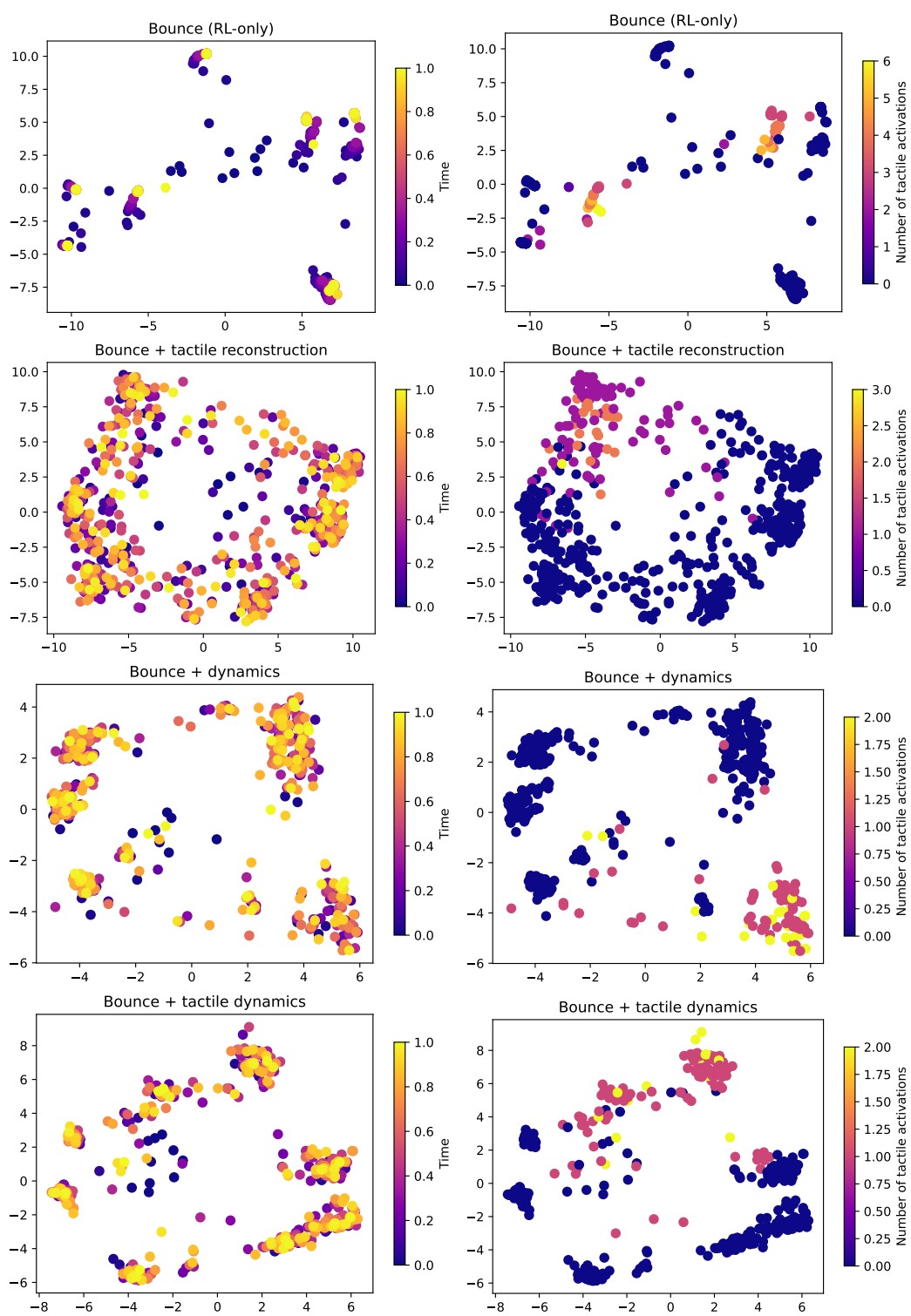

Figure A12: **Latent episode trajectory (PCA) of RL-only, TR, FD, and TFD *Bounce* agents.** *Left:* Samples colourised by time. *Right:* Samples colourised by summed binary contacts of the last tactile reading $\mathbf{o^{tact^t}}$.

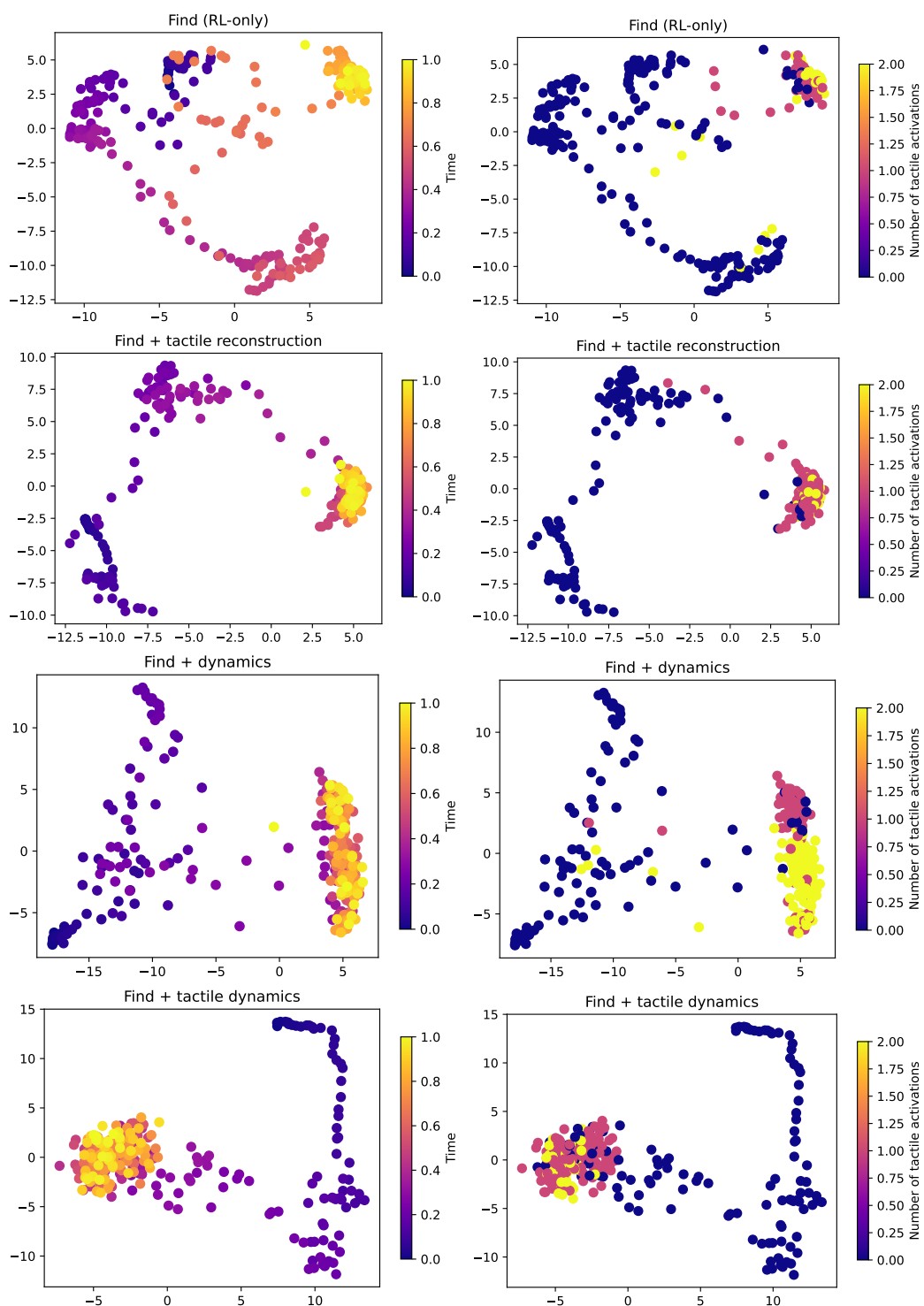

Figure A13: **Latent episode trajectory (PCA) of RL-only, TR, FD, and TFD *Find* agents.** *Left:* Samples colourised by time. *Right:* Samples colourised by summed binary contacts of the last tactile reading $\mathbf{o^{tact^t}}$.

