# OpenReview forum: "Enhancing Tactile-based Reinforcement Learning for Robotic Control"
_NeurIPS.cc/2025/Conference — NeurIPS 2025 poster_

### Official Review · Reviewer_Xsn5 · 2025-06-29

**Clarity:** 2
**Significance:** 3
**Originality:** 3
**Rating:** 5
**Confidence:** 3

**Summary:**

This work presents a new method for in-hand robot manipulation tasks, which introduces two dynamics objectives: tailored reconstruction and multi-step forward dynamics with auxiliary memory to address the issue of 1) the policy prematurely converging to features, 2) the difficulty of learning from sharp transitions and sparse nature data. The experiments show that the new methods outperform the previous work in multiple in-hand manipulation tasks.

**Questions:**

1. Are the two objectives mainly to help the training of the encoder? The decoder (d),  the forward model (f), and the predictor (p) are not used during inference ?
2. Are the two objectives designed for the encoder only? Not the value network and the policy?  If so, how can we prevent these two networks from falling into the problem in the hypotheses?
3. How does the tactile reconstruction prevent the model from "prematurely converging to features"? How does "Multi-step forward dynamics" help learning from sharp transitions and the sparse nature of tactile data more easily? For me, this seems to be hard to prove, but it is claimed in the paper that the new methods address the problems somehow. It is important to prove it.

**Ethical Concerns:**

["NO or VERY MINOR ethics concerns only"]

**Final Justification:**

The author addressed all my concerns and limitations during the discussion. I would like to accept the paper.

**Quality:**

3

**Strengths And Weaknesses:**

## Strengths

1. Tailored reconstruction and multi-step forward dynamics incorporate extracted learning objectives for the encoder, enabling the model to effectively capture and extract features from the observations, thereby facilitating the performance of downstream tasks.
2. The incorporation of the novel loss does not significantly enhance the complexity of the pipeline. These straightforward objectives maintain clarity and simplicity within the pipeline.

## Weakness

1. The experiments show that the overall performance increased, and the ablations show that the new objectives benefit the performance. But the proposed solutions for the hypotheses are not validated in the work.
    1. Tactile reconstruction softens the issue that learning "prematurely converges to features" is not discussed, as figures of value/policy loss show the gradients converging more smoothly.
    2. Also, no experiments show that "Multi-step forward dynamics" helps the learning from "sharp transitions and sparse nature

2. Figure 1 is very confusing, making it hard to trace the details of the main contribution.A more detailed caption should be added to the paper.  I am assuming
    1. the blocks/squares in the right two subfigures (ssl) from left to right are the feed-forward directions (even though they do not have arrows)
    2. SSL: tactile reconstruction: the upper sequence is a feed-forward network, and the lower arrow is the ground truth. L_BCE is computed by MSE from the upper and the lower
    3. SSL ulti-step forward dynamics: similar confusion to the above.

---

> ### Author Rebuttal · Authors · 2025-07-30
>
> We thank **Xsn5** for taking the time to review our paper and we appreciate the helpful feedback.
>
>
>
> **[Xsn5-1] This work “introduces two dynamics objectives: tailored reconstruction and multi-step forward dynamics with auxiliary memory.“**
>
> To clarify, we introduce three objectives for training tactile-based agents (Sec 3.2):
> 1. Tactile reconstruction: decode the tactile state from the multimodal representation through binary classification.
> 2. Full dynamics: multi-step forward dynamics on the multimodal representation.
> 3. Tactile dynamics: multi-step forward dynamics on the multimodal representation and decode tactile state from each predicted future representation.
>
> Tactile reconstruction (1) and tactile dynamics (3) are novel objectives, and the application of full dynamics (2) to tactile-based agents in model-free RL is novel.
>
>
>
> **[Xsn5-2] The proposed solutions for the hypotheses are not validated in the work… how does tactile reconstruction prevent premature convergence… how does dynamics help learn from sharp transitions?”**
>
> Our results (Figs 4, 5, 6) strongly demonstrate the efficacy of our methods, but we refrained from making definitive causal claims about the precise mechanisms underlying their success. For example, in Fig. 6 we observe that the vanilla PPO model significantly underperforms on the real-world metrics, perhaps lending some support to the theory that it has “converged prematurely”. In the same figure, we also observe that the addition of tactile signal and dynamics further improves performance on our benchmark tasks which feature “sharp transitions” between interaction states. We also attempted to provide additional insight into the learned representations through visualising future tactile states (Figs A5, A6) and performing latent trajectory analysis (Sec J) in the supplementary PDF.
>
> To further validate these hypotheses we can provide the following analysis in the revised paper. Note, the inclusion or omission of these results do not take away from our central contributions, but instead serve to provide additional insight into the unique challenges in this underexplored problem setting. Thanks for the suggestion!
>
> *H1: Preventing premature convergence through tactile reconstruction*
> * Hypothesis: The tactile reconstruction objective forces the network to maintain a meaningful representation of tactile input throughout training, preventing it from being “down-weighted” prematurely in favour of other modalities.
> * Experiment: We will track the L2-norm of the first layer weight vectors corresponding to proprioceptive and tactile inputs throughout training for each task and methodology. After normalising for input size, a larger contribution from the tactile input portion in agents trained with tactile reconstruction compared to RL-only agents will indicate that the model is extracting tactile features more effectively and from an earlier stage, preventing premature reliance on proprioception.
>
> *H2: Improving learning from discontinuous data through dynamics*
> * Hypothesis: By forcing the encoder to predict future state representations, the multi-step dynamics objective creates a more continuous learning signal, regularising learning from sparse and discontinuous tactile inputs.
> * Experiment: We will track the gradient norms of the input data with respect to the policy and value function. Firstly, we will establish if training with tactile data does indeed introduce greater gradient instabilities compared to proprioceptive agents. Secondly, we will evaluate if our dynamics-based objectives lead to enhanced optimisation (evidenced by reduced magnitude or frequency in gradient spikes). In addition, we will provide visualisations of the loss landscape of each agent to support our quantitative results on optimisation stability.
>
>
> **[Xsn5-3] Clarity of Fig. 1 and its caption.**
>
> We thank the Xsn5 for highlighting these clarity issues, and apologise if our figure was difficult to follow. In light of the comments, we will update Fig. 1 to resolve any potential ambiguities (e.g., adding missing arrows between components). We will add a legend to clarify that rectangular blocks denote feed-forward networks or layers, squares denote vectors, text denotes scalars, variables with “hats” denote model predictions, and those without are observed. Finally, we will update the caption to the following:
>
> *(Top left)*: Each timestep, proprioceptive (prop) and tactile (tact) readings are concatenated to form the timestep observation $o_t$. A stack of the $k$ last observations forms the current state $s_t$. The encoder $e$ learns the state representation $z_t=e(s_t)$ which the policy and value function are conditioned on. During learning, the self-supervised loss is added to the RL losses and backpropagated in a single pass. The  self-supervised loss optimises the encoder $e$ and its own task-specific networks ($d$, $f$, $p$).
> *(Top right)*: The tactile reconstruction objective optimises the encoder $e$ (and decoder $d$) to preserve the tactile state $s_t^{tact}$ in the representation $z_t$ by implementing reconstruction as binary classification with a binary cross-entropy loss $L_{BCE}$.
> *(Bottom right)*: The multi-step forward dynamics objective optimises the encoder $e$ (and forward model $f$ and projector $p$) to extract information from the state $s_t$ that will aid in predicting representations $n_f$ timesteps into the future.
> *(Bottom left)*: To further enhance learning, we propose to separate the RL memory and the auxiliary memory, which stores the last $N_{rollouts}$ of the RL memory.
>
> We confirm **Xsn5**'s understanding that the feed-forward direction of the blocks/squares on the right side of Fig. 1 is from left-to-right. To clarify, in tactile reconstruction $L_{BCE}$ is a binary cross-entropy loss (see Eqn 1) between the observed/ground truth tactile state (lower) and decoded/reconstructed tactile state (upper), not a MSE loss. We confirm that the dynamics loss is MSE (see Eqn 2), and we add that it is cumulatively summed for each timestep into the future: $L_1 + L_2 + … + L_{n_f}$. The computation of this loss is somewhat complex due to the need for a target network and projector (see L159-172 for full details), but we will strive to communicate this more clearly in the text and updated figure.
>
>
>
> **[Xsn5-4] Are the decoder, forward model, and the predictor used during inference?**
>
> These additional components are just used to assist with learning the encoder. Every network except the encoder and policy can be discarded for deployment.
>
>
>
> **[Xsn5-5] Potential fallibility of policy and value networks running into the same problem as encoder.**
>
> We confirm **Xsn5**’s understanding that yes, the self-supervised objectives are applied solely to the encoder (and relevant auxiliary networks). The policy and value networks are only optimised with the RL loss. The motivation of our approach is that through self-supervision, the encoder learns to generate a richer and more robust latent representation $z_t$ of the agent’s state. This “pre-processed” representation is then readily leverageable by the downstream policy and value function.
>
> Empirical evidence for this enhanced representation is In Sec J of the supplementary PDF, where we show the PCA reduction of the learned 256-dim state representation $z_t$ across an episode rollout. Taking Fig. A13 as the example, the latent trajectory of the self-supervised agents is much richer and diffuse between timesteps compared to the RL-only agents. This distinctive latent space provides evidence that self-supervision has a strong influence on the learned representation, allowing the policy and value networks to operate in a space less susceptible to the challenges of sensory data.

---

> > ### Comment · Reviewer_Xsn5 · 2025-08-06
> >
> > The rebuttal letter addressed all my concerns.
> > I would love to see the new experiments to valid the hypothesis [Xsn5-2], even if they should be clearly analyzed in the paper.
> > For now, the design experiments looks reasonable for me.

---

> > > ### Author Response · Authors · 2025-08-09
> > > **Author Response to Reviewer Xsn5**
> > >
> > > We appreciate **Xsn5**’s feedback. We have now performed some initial (one seed) experiments to test the two proposed hypotheses. We have re-run the original baselines with optimised hyperparameters, with and without self-supervised learning, to directly compare the loss of this signal. As we cannot provide plots/figures in the rebuttal, below we attempt to explain the overall trends.
> > >
> > > **H1: Tactile reconstruction preventing premature convergence to proprioceptive input**
> > >
> > > We tested this with two metrics, computed after each PPO update:
> > >
> > > 1.  **First layer weight norms:** For an input state size $[N_{prop} + N_{tactile}]$, the first weight matrix of the encoder is size $[1024, N_{prop} + N_{tactile}]$. We take the L2 norm for the tactile and proprioceptive neurons separately:
> > > * Tactile Weight Norm: $\lVert W_{tactile} \rVert_2 = \lVert W[1024, N_{tactile}] \rVert_2$
> > > * Proprioceptive Weight Norm: $\lVert W_{prop} \rVert_2 = \lVert W[1024, N_{prop}] \rVert_2$
> > >
> > > 2.  **Average Jacobian matrix norm:** For 10 randomly selected input states, we compute the Jacobian matrix of the learned representation $z_t$ with respect to the input state vector $s_t$ and average the results. The Jacobian matrix is size $[256, N_{prop} + N_{tactile}]$, and we follow the same process of computing norms for the tactile and proprioceptive parts.
> > >
> > >
> > > For all training timesteps in all environments, the tactile reconstruction objective results in a larger weight norm for the tactile input. The Jacobian analysis shows a similar pattern:  the tactile input has a more significant influence on the final representation $z$ when the reconstruction objective is active. For the camera-ready version, we will include a thorough multi-seed analysis of each objective/environment combination.
> > >
> > >
> > > **H2: Improving learning from discontinuous data through dynamics**
> > >
> > > To interpret the stability of our optimisation process, we computed the L2 gradient norms of the policy and value losses with respect to the learned representation $z$, $\lVert \nabla{z} L_{\pi} \rVert_2$ and $\lVert \nabla{z} L_{V} \rVert_2$, respectively. These metrics were computed and averaged per minibatch during the PPO update.
> > >
> > > The results varied by environment:
> > >
> > > * In Find, the dynamics objective leads to a lower magnitude and oscillation amplitude for both $\lVert \nabla_{z} L_{V} \rVert_2$ , $\lVert \nabla{z} L_{\pi} \rVert_2$ across all timesteps.
> > > * In Bounce and Baoding, $\lVert \nabla{z} L_{\pi} \rVert_2$ is much higher (same amplitude). For $\lVert \nabla_{z} L_{V} \rVert_2$, we observed a higher peak coinciding with greater agent returns, followed by convergence to the same level as the baseline PPO agent.
> > >
> > > The results suggest that the dynamics objective affects the latent representation differently depending on the task's requirements. In Find, the lower gradient norms may indicate that the dynamics objective helps create a more robust and less volatile representation, perhaps helping the agent synthesise the sudden appearance of tactile input. In Bounce and Baoding, the higher policy gradient norm may indicate the dynamics objective enables the policy to be more sensitive and reactive to fine-grained changes in the latent state, which is beneficial for tasks demanding precise, high-frequency control. The higher initial peak in the value gradient norm indicates a faster rate of learning and adaptation to new, higher-return behaviors. For the camera-ready version, we will include a thorough multi-seed analysis of each objective/environment combination.

---

> ### Comment · Area_Chair_G4vz · 2025-08-05
> **Please respond to authors**
>
> Dear reviewer Xsn5,
>
> Thank you for your reviewing efforts so far. Please engage with the authors' response.
>
> Thanks,
> Your AC

---

### Official Review · Reviewer_WBY5 · 2025-07-03

**Clarity:** 4
**Significance:** 3
**Originality:** 3
**Rating:** 5
**Confidence:** 4

**Summary:**

This work focuses on enhancing tactile-based (RL) for robotic control by addressing the challenge of effectively exploiting sparse tactile sensory signals. The paper proposes training of RL policies with a binary reconstruction of tactile data streams via an encoder architecture, hypothesizing the higher likelihood of signal value and sim2real transfer over continuous force readings. They further propose introduction of auxiliary reconstruction and prediction objectives and separate memory for the RL task policy and auxiliary objective policy in order to encourage the policies not to ignore the haptic signals in favor of richer proprioceptive signals. To test the hypotheses and validate their claims, the robot tactile olympiad simulation benchmark is introduced and results demonstrate the super-human performance on bounce and baoding tasks in simulation.

**Questions:**

I am interested in the authors' opinions on the value of pursuing and optimizing for superhuman results in simulation. I think it is great to see such results and the videos help to put the quantitative analysis into context, but I know first-hand how simulation inaccuracies can enable non-physical behaviors which policies can and will latch onto. For example, oscillatory impulses applied by contact resolution between two frames at high frequency could masquerade as "bouncing" or "vibration". Concretely, imagine how different soft contacts between a human hand and ball would be from a rigid approximation in an impulse based engine. The question boils down to: "how far should we trust impulse-based simulation models with regards to highly dynamic tasks?".

**Ethical Concerns:**

["NO or VERY MINOR ethics concerns only"]

**Final Justification:**

During the rebuttal phase the authors have answered my and fellow reviewers questions adequately to keep my continued support for the acceptance of this work.

**Limitations:**

yes

**Paper Formatting Concerns:**

Typos:

LN 75: "that that"
LN 244: "that that"
LN 292: "the some of the"
LN 285: "also also"
supp D.2 "underactuacted"

**Quality:**

3

**Strengths And Weaknesses:**

Overall interesting hypothesis, well conducted research, and strong results/discussion. The auxiliary objectives, novel architectures, and new benchmark appear to be positive contributions to the field and I appreciate the focus on scientific discussion and suggestions to others working on similar problems. Of course it would be great to see sim2real results and any result obtained in a single impulse-based simulation on dynamic interactive tasks must be taken with a grain of salt. That said, I see clear impact and novelty here and support acceptance of the work.

Pros:
- novel formulation
- auxiliary objectives to force the policy to encode the haptic data
  - future state prediction is a good objective to encourage learning from sparse signals
- separate memory for auxiliary haptic objectives allow for optimization on a wider distribution of data offline
- significant improvements in capability on test tasks
  - simulated agent is superhuman in bounce and baoding
- robot tactile olympiad dataset
- binary tactile data does appear to be more transferable between sim and real in the short term
- Good hyper-paramter tuning to ensure validity of results
- Nice focus on and interesting format of discussion section adds scientific value to the work
  - I like the "translation to practical recommendations" as a focus on advancing the field

Cons:
- binary contact signals are limiting for more practical tasks
- no real world transfer experiments
  - because physics simulation is impulse based approximation, it isn't clear if superhuman performance would be possible in the real world vs. simulation
- large variances for complex task policies
- No unified hyper-paramter setting for all tasks indicates challenges in generalization

---

> ### Author Rebuttal · Authors · 2025-07-30
>
> We thank **WBY5** for their careful review and for the highly insightful comments!
>
> **[WBY5-1] Are the superhuman policies realistic?**
>
> We agree that any results for a dynamic contact-rich task in simulation must be critically examined.  Based on our analysis, we do not believe our superhuman performance is due to exploiting simulation inaccuracies. We have several pieces of evidence for this:
>
> 1. *Oscillatory impulses masquerading as bouncing*: The maximum bounce count was 88 over a 600-timestep episode, which translates to approximately seven timesteps between each bounce. By slowing down the provided videos to 0.25x speed, it is evident that the SSL-optimised agent has learned a sophisticated strategy involving both distinct bounces (see timestamp 00:09) and a secondary, vibration-like behaviour (see timestamp 00:07), which together enable the superhuman performance. This strategy contrasts with the RL-only agent’s repetitive and less effective motions, suggesting our method has learned a superior behaviour rather than exploiting a simulation artifact.
> 2. *Similar improvements in static tasks*: Our method also shows significant performance improvements in the “Find” task, which is a completely static, non-dynamic environment. In this task, our agents found the target object on average 1.5 seconds faster than the baselines (see Fig. A1 in the supplementary PDF). This demonstrates that our method's benefits are not exclusive to dynamic environments, reinforcing the assertion that our core contribution is a more effective learning strategy rather than the exploitation of dynamic physics.
>
>
> **[WBY5-2] Is there value in optimising for superhuman performance in simulation?**
>
> We believe there is great value. See L275 to 278 for a discussion of our performance in the context of human baselines. A key tenet of modern reinforcement learning research, from Atari to AlphaGo, has been to push beyond human capabilities in simulation as a means of discovering novel and superior strategies. In our work, we were initially focused on the fundamental challenge of getting a "blind" agent to perform any meaningful action at all. The discovery that our methodology enables superhuman capabilities was an unexpected but powerful outcome, attesting to its ability to unlock creative solutions for challenging, partially observable control tasks.
>
>
>
> **[WBY5-3] What is the likelihood of sim2real transfer?**
>
> We believe that the benefits of our method should translate to the real world, assuming a successful sim2real gap can be bridged for the base tactile agent. This is because our core contribution is a method for improving the learning process and generating superior policies. If the physical assumptions are generally valid, these strategies should be transferable. Our deliberate choice of binary contact signals and a methodology that only modifies network weights were made to specifically facilitate this future transfer. Please see response **[gp4F-4]** for more discussion of sim2real.
>
>
>
> **[WBY5-4] Binary contact signals are limiting for more practical tasks.**
>
> The choice of binary signals is a deliberate one, driven by practical constraints and the state of sim2real research. While richer sensory data is theoretically advantageous, it incurs substantial computational costs (e.g., see discussion on L305) and faces a significant sim2real gap, even for continuous signals due to the difficulty of accurately simulating contact physics.
>
> Binary tactile sensors, in contrast, are cheap, robust, and require minimal calibration (e.g. can employ a low pass filter and then threshold the voltage [A]). Our work demonstrates, for the first time, that it is possible to achieve superhuman performance using only these simple measurements. This finding not only challenges the perception that binary signals are limiting but also establishes a highly feasible and effective baseline for real-world tactile learning.
>
> [A] Lee et al., DexTouch: Learning to Seek and Manipulate Objects with Tactile Dexterity, IEEE RAL
>
> **[WBY5-5] Large variances for complex task policies.**
>
> There is indeed a large variance observed in the policies for the complex Baoding task (see Fig. 4). We attribute this variance to the difficult nature of this partially observable control task and the relatively sparse reward signal (Sec D3 in the supplementary PDF). A single successful rotation provides a large, sparse bonus (+10 compared to distance reward within [0, 0.1]), making agents vulnerable to this high-variance behavior.
> The ability to mitigate this challenge is a key contribution of our work. As shown in Fig. A3 (bottom left) of the supplementary PDF, the RL-only agents and the full dynamics agents both exhibit significant performance variance, with multiple seeds failing to achieve a successful rotation. However, our tactile reconstruction agent (i.e., “+tactile recon”) is the only method that achieves consistent, low-variance returns, with all seeds successfully learning the task. The ability of our approach to significantly reduce variance in this difficult setting is a notable and primary result.
>
> **[WBY5-6] No unified hyper-parameter setting for all tasks indicates challenges in generalization.**
>
> This is an interesting question. However, we believe that diverse hyperparameter settings across tasks are a necessary and a sound methodological choice. The tasks in our study (see Fig. 2) are intentionally diverse, encompassing distinct contact dynamics, devices,  and control objectives. As is standard practice in reinforcement learning research, it is highly unlikely that a single set of hyperparameters would be optimal across such a varied set of problems. Thus, we followed the best practice (e.g., as in [B]) of performing a hyperparameter search for each method on each task.
>
> This approach is crucial for validating the true performance of our method. The fact that our methods consistently outperform the baselines after both are optimally tuned for each specific task provides stronger evidence for its efficacy. We will release our training code, including the hyperparameter search configurations and final values to facilitate reproducibility and future research. We will also clarify the hyperparameter settings in the revised text.
>
> [B] Eimer et al., “Hyperparameters in Reinforcement Learning and How To Tune Them”, ICML 2023

---

> > ### Comment · Reviewer_WBY5 · 2025-08-06
> >
> > Thank you, authors, for your responses and discussions around my questions and concerns. In short, I remain supportive of publishing this work and my prior "accept" rating remains the same.
> >
> > 1) I'm glad to hear that over-indexing on simulation artifacts was considered and I agree with your assertion that improvement on non-dynamic tasks and improvement over baselines in dynamic tasks are significant signals in favor of the effectiveness of the proposed learning strategies. However, I would be remiss not to raise this for discussion.
> >
> > 2) I should clarify here that I completely support pursuit of super human policies. My real question is regarding the value of pursuing this performance level in simulation alone and on toy problems like baoding vs. general purpose tasks such as dexterous manipulation of household objects or tool use. In the specific case of this paper, I agree that showing super human performance in simulation is promising result worth sharing.
> >
> > 3) Agreed that binary contact signals are more likely to transfer to real world. Also, I acknowledge that the question doesn't really have a good answer. We can't know whether things will transfer until we try as other reviewers have also pointed out. I personally don't think this should be a blocker to publication, but I sincerely hope that transfer experiments are in the pipeline.
> >
> > 4) Fair enough. The potential upper bound of value for continuous signals seems much higher, but proving the value of binary signals would allow us to make progress now. I support this conceptually.
> >
> > 5) Thank you for this clarifications here, my understanding is updated. I'll first acknowledge my own mistake here: I conflated some of the figure results initially. I originally interpreted the large training variances (e.g. of Figure 3 PPO(prop-tactile)) incorrectly. I agree with your assertion that the tactile reconstruction agent's low variance indicates escape from local minima encountered by other variations which supports the efficacy of the proposed approach.
> >
> > 6) Fair enough. Agreed that hyper-parameter search is standard and that if baselines had equal opportunity then the gains are legitimate. I appreciate any clarifications of concrete details such as hyper-paramter settings in the final paper as they will aid with reproducibility and validation by others. That said, I maintain my opinion that reducing reliance on hyper-parameter search would make the general tool of standard RL more valuable as task complexity and computational cost increases. Clearly addressing this general criticism of standard RL application isn't in the scope of the current work.

---

> > > ### Author Response · Authors · 2025-08-08
> > > **Author Response to Reviewer WBY5**
> > >
> > > 1. We are pleased you raised this interesting point, and consequently will include a short discussion in the camera-ready version.
> > > 2. We think that there is value in pursuing superhuman performance in simulation, even if we venture into the realm where transfer becomes too difficult. This is because it may lead to the discovery of more powerful algorithms, that are also inherently benefical to simpler, general-purpose tasks.
> > > 3. Yes, we would like to clarify that our ultimate interest lies in real-world robotics, and our current focus now is sim2real.
> > > 4. We agree that continuous signals likely offer a higher upperbound of performance. After establishing sim2real for binary, we will work on adding increased sensory information. Interestingly, the developers of Isaac Lab have stated that the continuous contact forces have been incorrect in all releases up until now (https://github.com/isaac-sim/IsaacLab/releases/tag/v2.1.1), possibly explaining some of the cited transfer difficulties.
> > > 5. We are glad we could help clarify this point.
> > > 6. We completely agree that reducing reliance on hyperparameter search (and also multiple seeds) would make RL significantly more valuable and accessible. As we detail in Section 4, sweeping alone consumed  80% of the computational resources for this paper. We hope that this extensive effort provides the community with confidence in the validity of our results.
> > >
> > > We would like to thank reviewer WBY5 for their thoughtful feedback and continued engagement with our work throughout the rebuttal period.
> > >
> > > Best regards,
> > >
> > > Authors

---

> ### Comment · Area_Chair_G4vz · 2025-08-05
> **Please respond to the authors' response**
>
> Dear reviewer WBY5,
>
> Thanks for your reviewing efforts so far. Please respond to the authors' response.
>
> Thanks,
> Your AC

---

### Official Review · Reviewer_gp4F · 2025-07-03

**Clarity:** 3
**Significance:** 3
**Originality:** 3
**Rating:** 5
**Confidence:** 4

**Summary:**

This paper posits that the current tactile manipulation policies entangle the proprocioptive and tactile representations making it unclear and also hard to attribute the benefit of "tactile" in tactile manipulation. To investigate this, the authors propose to disentangle and learn SSL methods on top of the tactile states as a form of auxiliary loss to guide the RL policy learning. Specifically, they consider tactile dynamics and tactile reconstruction as 2 SSL tasks.
Further, to evaluate the method, the authors propose a benchmark - RoTO: Robot Tactile Olympiad that includes 3 tasks, namely Find, Bounce, and Baoding involving parallel gripper (franka arm) and dextrous manipulation (shadow arm).

**Questions:**

3. Figure 1 (right) -- the architecture for multi-step forward dynamics is confusing since there are both L1 and L2 losses being taken. I presume that this was a typo and it's L2 loss everywhere (based on the multi-step forward dynamics subsection). But I wanted to confirm if my assumption is correct.

4. For the forward dynamics, is there any prioritized sampling performed to reduce the imbalance present in the tactile state transitions?

5. Can the authors show the plots for the ablation study on $n_f$ for at least one task? I'm curious how the values were chosen. Was the last $n_f$ value that didn't result in a significant increase from $n_f-1$ selected as the optimal value?

**Ethical Concerns:**

["NO or VERY MINOR ethics concerns only"]

**Final Justification:**

Thanks to the authors for their comprehensive answers during the rebuttal phase. I have looked over all the comments and believe that the authors have adequately addressed all my major concerns.

Thanks for clarifying the summary of the paper as well. My major concerns were lack of sim2real experiments and related works. While I still am of the belief that experiments in sim, especially in the RL-simulation community are not trivially transferrable to real world (sorry for the typo in my comment where I said "correlation between simulation experiments working versus the policy actually working on the real robot is quite large" when I actually meant that the correlation is quite low) -- I do think that the methodological contributions of this work are well suited for this conference.

Hence, I will increase my score to **Accept**.

**Limitations:**

Yes.

**Paper Formatting Concerns:**

None.

**Quality:**

3

**Strengths And Weaknesses:**

**Strengths**

- Well-written paper

- The idea of specifically learning tactile dynamics is an interesting one and a good contribution to the field of tactile manipulation.

- Thorough hyperparameter search including learning rate(!), which is often missing in a lot of RL papers.


**Weaknesses**


1. Related works miss several tactile-based dynamics models such as RoboPack [1] and Swing Bot[2]. Citing & discussing them would be helpful for the reader.

2. Lack of Sim2Real experiments. This is acknowledged by the authors, however, I must note that the correlation between simulation experiments working versus the policy actually working on the real robot is quite large, especially for dexterous manipulation, hence I do consider this limitation as a major drawback of the work.

---

**References**

[1] RoboPack: Learning Tactile-Informed Dynamics Models for Dense Packing, Bo Ai et al., RSS 2020.

[2]SwingBot: Learning Physical Features from In-hand Tactile Exploration for Dynamic Swing-up Manipulation, Chen Wang et al., IROS 2020

---

> ### Author Rebuttal · Authors · 2025-07-30
>
> We thank **gp4F** for taking the time to review our paper and for the great questions.
>
>
> **[gp4F-1] Paper summary and the entanglement of proprioceptive and tactile representations.**
>
> We appreciate **gp4F**’s summary of our work, but we would like to clarify a key potential misunderstanding. The summary states that we posit entanglement of proprioceptive and tactile representations as the problem. However, our premise is precisely the opposite, i.e., we hypothesise that *the tactile observation often fails to adequately entangle or contribute meaningfully to the overall multimodal representation* (see discussion on L31-40). Thus, the challenge we address is not the difficulty in attributing the benefit of tactile input, but rather the observed lack of expected performance improvement when tactile observations are concatenated to the observation.
>
> Furthermore, while our tactile dynamics and tactile reconstruction objectives are designed to encourage the tactile state to be more salient within the multimodal representation, the key finding was that the most successful approach was the multi-step forward dynamics, which we denoted as “full dynamics” (see results in Fig 4).
>
>
>
> **[gp4F-2] Thorough hyperparameter search is performed.**
>
> We appreciate **gp4F**’s recognition of our thorough hyperparameter search. We fully agree that the common omission of such rigorous tuning in many RL for robotics works can undermine the reliability of reported results. To help address this challenge, we plan to release our integrated hyperparameter optimisation code with Isaac Lab (where it is not available) alongside our benchmark, which we hope will encourage and facilitate the adoption of best practices within the research community.
>
>
>
> **[gp4F-3] Additional related works.**
>
> We thank **gp4F** for highlighting the omission of these two tactile-based dynamics works (RoboPack and SwingBot). Our related work section primarily focuses on model-free RL with self-supervised learning. We had originally omitted SwingBot because the focus of that paper is about generalisation to unseen physical properties and it does not use RL, whereas we focus only on in-domain expertise with RL. Similarly, RoboPack uses trajectory optimisation after the dynamics model is learned, whereas we explore improving online policy optimisation with online dynamics learning. However, we appreciate that a discussion of achievements in learning tactile dynamics within other communities, particularly model-based policy learning (e.g., deep tactile MPC [A]), would indeed be valuable and provide a more comprehensive context for the reader. We will revise our related work section to include a discussion of these additional references.
>
> [A] Tian et al., Manipulation by Feel: Touch-Based Control with Deep Predictive Models, ICRA 2019
>
> **[gp4F-4] Lack of sim2real experiments on real robots for dexterous manipulation.**
>
> We acknowledge this point regarding the absence of sim2real experiments, which was also a focus of some of our discussion with reviewer **WBY5** (please see response **[WBY5-[1-3]]** for further context). We believe the absence of real-world experiments does not detract from our work’s core contributions for the following reasons:
>
> 1. *Establishing a simulation baseline for novel tactile-based robotics.* We address an important but largely unexplored domain, i..e, tactile-based robotic agents operating without visual input. Given the scarcity of prior work (see L41-50), our paper provides a crucial foundational benchmark for future research in this novel setting.
>
> 2. *Solving a fundamental problem of neural network-based RL agents.* Our paper's central contribution is a new methodology that markedly improves how agents leverage binary tactile observations in simulation (see Fig. 6). As demonstrated in the supplementary videos and latent trajectory analysis in Sec J of the supplementary PDF, agents using our method discover unique, effective strategies for complex, partially observable control tasks that traditional RL agents do not.
>
> 3. *Robustness across diverse dynamics.* To counter concerns about simulation-specific gains, we validated our method across three tasks with distinct contact dynamics and different simulated robotics hardware (Find: sparse, Bounce: intermittent, Baoding: continuous). The consistent high performance across these varied settings and robot morphologies (see Figs. 4 and 5), strongly indicates a genuine advancement in learning capabilities, rather than a mere simulation artifact.
>
> 4. *Designed for sim2real transfer*. Our design choices, particularly the use of binary contact activations, are deliberate steps to facilitate future real-world transfer. Prior work [61, 63] has demonstrated robust sim2real transfer with binary tactile activations even in dexterous manipulation tasks. Furthermore, our method's focus solely on modifying network weights simplifies real-world deployment. We contend that a superior simulated policy, assuming the sim2real gap is otherwise bridged for a tactile-based RL agent, should indeed transfer, and our validation across diverse tasks suggests our policies are not unduly reliant on unrealistic simulation physics as asked by **WBY5**.
>
> We believe that our comprehensive empirical study in controlled simulated environments provides substantial findings and novel insights that will be of immediate interest and benefit to researchers working at the intersection of machine learning and robotics. Specifically, we demonstrate the proven efficacy of binary tactile activations for complex dexterous manipulation, addressing a key open question in the tactile sensing community. We posit that it is important to first establish the efficacy of our findings in simulation, via a repeatable benchmark that can be used by others, before moving on to real world robotics experiments in future work.
>
>
> **[gp4F-5] Losses depicted in Fig 1.**
>
> We apologise for the confusion caused by the notation in Figure 1 (right) regarding the multi-step forward dynamics. **gp4F**'s assumption about it being a typo is understandable. To clarify, the subscripts L1​, L2​,…​ do not refer to different types of loss (e.g., L1-norm vs. L2-norm), but rather denote the loss calculated for the prediction at the first step (i.e., L1​), second step (i.e., L2​), and subsequent steps into the future from the current state. The specific loss function used for all these steps is consistently the L2 loss, as correctly identified by the reviewer and detailed in the multi-step forward dynamics subsection. We recognise that this notation is ambiguous and will revise Figure 1 and its caption (see response **[Xsn5-3]**) in the camera-ready version to explicitly clarify this.
>
>
> **[gp4F-6] Is any prioritized sampling used to reduce the imbalance present in the tactile state transitions for the forward dynamics model?**
>
> This is an interesting question. While we did employ positive weighting for tactile reconstruction to address sparsity (see L288), we confirm that *no* prioritised sampling mechanism was used for the dynamics learning. We will clarify this in the revised text.
>
> During initial investigations, we experimented with various sampling approaches, such as rule-based strategies and exponential weighting. However, we did not observe significant performance gains in our specific experimental setup. We wish to emphasise that this outcome does not diminish the potential value of prioritised sampling in tactile dynamics learning. On the contrary, we think that it might be a critical component for handling tactile sparsity and imbalance. The exploration of prioritised sampling, particularly in conjunction with a separated auxiliary memory for transitions, represents a potentially interesting direction for future research.
>
>
> **[gp4F-7] Can the authors perform an ablation study on $n_f$?**
>
> We agree with **gp4F** that the impact of the number of forward prediction steps, $n_f​$, is highly interesting and critical for performance. To ensure robustness and account for potential interactions with other hyperparameters (e.g., learning rate), we treated $n_f​$ as an optimized hyperparameter within our search space, sweeping over values of {2, 3, 4, 10}. This approach is detailed in Appendix I, Table A3, and we will ensure this optimisation strategy is clearly stated in the main paper body.
>
> Our findings revealed task-specific optimal values for $n_f​$. For the 'Find' and 'Baoding' tasks, optimal performance was achieved with $n_f$​ values between [2, 4]. Interestingly, for the 'Bounce' task, a longer sequence length of $n_f​=10$ proved most effective, likely due to its ability to capture a full bounce trajectory, as illustrated in Fig. A5.

---

> ### Comment · Area_Chair_G4vz · 2025-08-05
> **Please respond to the authors' response**
>
> Dear reviewer gp4F,
>
> Thanks for your reviewing efforts so far. Please respond to the authors' response.
>
> Thanks,
> Your AC

---

### Official Review · Reviewer_zxcc · 2025-07-05

**Clarity:** 2
**Significance:** 2
**Originality:** 2
**Rating:** 2
**Confidence:** 4

**Summary:**

This paper proposes a general-purpose self-supervised learning methodology that effectively leverages tactile input for downstream robotic control tasks in RL. Firstly, the manuscript provides a novel application and analysis of tailored reconstruction and multi-step dynamics objectives that enable the agent to leverage its tactile observations. Secondly, the proposed method proposes to separate the auxiliary memory from the RL rollout memory to stabilise and strength the self-supervision updates. Experimental results show the superiority of the proposed self-supervised agents compared to RL-only agents on three kinds of tasks: Find, Bounce, Baoding in Issac Lab.

**Questions:**

Please see Weaknesses section. In my opinion, the poor organization of the manuscript, the lack of real-world experiments, and the insufficient method design are the major issues of the manuscript. Therefore, I lean to recommend rejection in the current version of submission.

**Ethical Concerns:**

["NO or VERY MINOR ethics concerns only"]

**Final Justification:**

I appreciate the authors for providing the rebuttal, which addresses some unclear points, such as the training objectives and missing implementation details. However, my concerns about the lack of technical contribution and sim2real transfer experiments are not well-solved through the rebuttal. The lack of real-world experiments are also indicated by many other reviewers. Based on the above concerns, I maintain my original score.

**Limitations:**

Yes

**Quality:**

2

**Strengths And Weaknesses:**

### Strengths
1. The proposed method show the superiority over RL-only agents on multiple kinds of tactile-based complex manipulation tasks.

2. The authors provide various aspects of discussion to analyze the advantage of the proposed dynamics objective and self-supervised learning strategy.

3. Supplementary materials are provided with detailed implementation details and additional experimental results.

### Weaknesses
1. A major concern is the organization and presentation of the manuscript, which largely impact the readability. For example, the Method section contains a lot of symbols without providing a notation summary table to explain the meaning of each variable. Moreover, the captions of figures do not explain the figure (e.g., the pipeline of method and the meaning of the symbols in Figure 1) in detail. Thirdly, what is the total training objective and how the self-supervised learning cooperate with the RL objective are unclear from Figure 1. Fourthly, In the experimental results, the authors only provide the comparison results in the form of figures instead of tables, making the comparison indirect. I strongly suggest the authors to refine and reorganize their manuscript to improve the clarification and readability.

2. The design of the proposed method seems to lack enough technical details and contents. The core contribution is the self-supervised training strategy, which has been widely adopted in previous works. The separate auxiliary memory is also an important contribution as described in Line 77-80. However, the authors only introduce it briefly in Section 3.3.

3. In Line 70, the authors claim that the proposed method is supposed to have low sim2real gap, however, the manuscript does not contain the real-world experiments, as indicated in the Limitation section. These makes the adaptability of the proposed method to the real world doubtful.

---

> ### Author Rebuttal · Authors · 2025-07-30
>
> We thank **zxcc** for their comments and constructive feedback. As a reminder, we introduce a new approach that improves sensory-driven robotic agents for complex manipulation tasks that does not make idealised assumptions about the quality of tactile state information available. In addition, we introduce three new environments for benchmarking tactile-based manipulation methods.
>
> **zxcc**’s concerns primarily relate to the structure of the paper, missing details, and the lack of experiments on real robots. With the exception of the last one, which we attempt to justify why its absence does not impact our contributions, the other concerns can be addressed by updating the final camera ready text.
>
>
> **[zxcc-1] Organization and presentation of the paper.**
>
> We thank the reviewer for highlighting these clarity issues, and will make the following changes in light of zxcc’s comments:
> * *Notation:* While all symbols are sequentially introduced in the text, we recognise that a central reference would be helpful. We will add a summary of the most important notation at the beginning of the Method section to serve as a quick reference for the reader.
> * *Figure 1 caption:* We will update the caption for Fig. 1 (see response **[Xsn5-3]**) so that it is more self-contained and will expand it to provide a more detailed step-by-step description of our method's pipeline in the camera-ready version.
>
>
> **[zxcc-2] Training objective and role of self-supervised learning.**
>
> The total training objective is given in L147: _“The self-supervised loss is added to the policy, value, and entropy loss and backpropagated in a single pass. The auxiliary loss optimises the encoder e and its own task-specific networks.”_ We did not include this in the original figure because it is the default optimisation strategy in self-supervised RL, and is not a novel contribution for our work. In the camera-ready version, we can explicitly add the loss summation for improved clarity.
>
>
> **[zxcc-3] Quantitative results are presented as figure instead of table.**
>
> In the main body of the paper we chose to display results as a figure of the agent’s mean evaluation return throughout learning, because we are interested in understanding the impact of incorporating self-supervised objectives throughout the optimisation process, and figures provide more information on sample efficiency and stability. For a more direct comparison, we provided a bar chart in Fig. 6 to show how different methodologies correspond to improvements in physical quantities. In the camera-ready version, we can add a table of the highest obtained mean evaluation returns for each methodology.
>
>
> **[zxcc-4] Missing implementation details about the proposed method.**
>
> Our problem setting is outlined in Sec 3.1, the self-supervised objectives in Sec 3.2, and auxiliary memory is described in Sec 3.3. Additional implementation details are provided in Sec 4 and Sec E, F, and I of the supplementary PDF. We will revise the descriptions and more clearly point to the additional implementation details in the main paper. If there are specific aspects of the implementation details that are not clear, we are happy to discuss them during the author-reviewer discussion period - please do not hesitate to ask additional questions.
>
> **[zxcc-5] Self-supervised training has been used in previous works**
>
> Yes, we agree, and do not claim in the paper that the use of self-supervised learning in combination with reinforcement learning is novel. In the related work (Sec 2) we summarise how different self-supervised training strategies have been adopted over the past decade. Indeed, reconstruction and dynamics-based objectives have been widely adopted in previous works, because of their general-purpose nature. Please see the summary of our contributions in the introduction (Sec 1) and response [zxcc-7] for the novelties we claim.
>
>
>
> **[zxcc-6] The separate auxiliary memory is an important contribution but only briefly discussed.**
>
> We thank **zxcc** for highlighting the separate auxiliary memory as an important contribution. The description is brief because the method itself is intentionally simple, and we have described the implementation in its entirety in Sec 3.3. The contribution is not an unnecessarily complex new algorithm, but rather the insight that decoupling the auxiliary memory from the on-policy buffer and significantly expanding its size can be a simple yet powerful technique (see results in Fig. 5).
>
> To be concrete, for an agent using $B$ parallel environments and a rollout length of $R$, the on-policy data has a shape of $[B, R, ...]$. Our method simply involves storing data for the auxiliary tasks in a larger, separate buffer of size $[N_{rollouts}, B, R, ...]$, where $N$ is the number of past rollouts we keep in memory.
>
> **[zxcc-7] Sim2real gap and real world experiments.**
>
> We wish to clarify that our paper does not claim the proposed method has a low sim2real gap; rather that we designed our methodology with this long-term goal in mind. As stated in our Introduction (L62), _“We desire the following characteristics… (d) low sim2real gap: inspired by the low-cost setups in [61, 63], we study learning with binary tactile activations… to avoid transfer difficulties that can come with continuous measurements [61]”_. We also acknowledge in our Limitations (L314) the absence of real-world experiments and express we _"hope our choice to study binary tactile activations greatly minimises the potential sim2real gap”._
>
> We believe the absence of real-world experiments does not detract from our work’s core contributions for the following reasons:
>
> 1. *Establishing a simulation baseline for novel tactile-based robotics.* We address an important but largely unexplored domain, i..e, tactile-based robotic agents operating without visual input. Given the scarcity of prior work (see L41-50), our paper provides a crucial foundational benchmark for future research in this novel setting.
> 2. *Solving a fundamental problem of neural network-based RL agents.* Our paper's central contribution is a new methodology that markedly improves how agents leverage binary tactile observations in simulation (see Fig. 6). As demonstrated in the supplementary videos and latent trajectory analysis in Sec J of the supplementary PDF, agents using our method discover unique, effective strategies for complex, partially observable control tasks that traditional RL agents do not.
> 3. *Robustness across diverse dynamics.* To counter concerns about simulation-specific gains, we validated our method across three tasks with distinct contact dynamics and different simulated robotics hardware (Find: sparse, Bounce: intermittent, Baoding: continuous). The consistent high performance across these varied settings and robot morphologies (see Figs. 4 and 5), strongly indicates a genuine advancement in learning capabilities, rather than a mere simulation artifact.
> 4. *Designed for sim2real transfer*. Our design choices, particularly the use of binary contact activations, are deliberate steps to facilitate future real-world transfer. Prior work [61, 63] has demonstrated robust sim2real transfer with binary tactile activations even in dexterous manipulation tasks. Furthermore, our method's focus solely on modifying network weights simplifies real-world deployment. We contend that a superior simulated policy, assuming the sim2real gap is otherwise bridged for a tactile-based RL agent, should indeed transfer, and our validation across diverse tasks suggests our policies are not unduly reliant on unrealistic simulation physics as asked by WBY5.
>
> We believe that our comprehensive empirical study in controlled simulated environments provides substantial findings and novel insights that will be of immediate interest and benefit to researchers working at the intersection of machine learning and robotics. Specifically, we demonstrate the proven efficacy of binary tactile activations for complex dexterous manipulation, addressing a key open question in the tactile sensing community. We posit that it is important to first establish the efficacy of our findings in simulation, via a repeatable benchmark that can be used by others, before moving on to real world robotics experiments in future work.

---

> > ### Comment · Reviewer_zxcc · 2025-08-05
> >
> > Thanks the authors for providing the rebuttal to address some unclear points like the training objectives and missing implementation details. However, my concerns about the lack of technical contribution and sim2real transfer experiments are not well-solved through the rebuttal. The lack of real-world experiments are also indicated by many other reviewers. Based on the above concerns, I decided to maintain my original score and I suggest a thorough improvement of the manuscript according to the reviewers' suggestions.

---

> > > ### Author Response · Authors · 2025-08-05
> > > **Author Response to Reviewer zxcc**
> > >
> > > **Technical contribution**
> > >
> > > We respectfully disagree with the assessment that our work lacks a technical contribution. As detailed in our rebuttal (see response [zxcc-5,7]) and manuscript (L73-87), our primary contribution is a novel application and analysis of SSL training objectives to enhance tactile-based robotic control. In the context of model-free SSL+RL, we are the first to:
> > >
> > > - Reconstruct only the tactile observation from the multimodal representation.
> > > - Formulate tactile reconstruction as a binary classification task.
> > > - Apply multi-step forward dynamics to tactile data.
> > > - Provide evidence for the benefits of training on off-policy data.
> > > - Conduct a comprehensive analysis of the effect of different SSL objectives on tactile-based agents.
> > >
> > > We maintain that, given the challenge, importance, and novelty of tactile-based RL, these are valuable contributions in their own right, and an important step to take prior to full-scale evaluation on a real-world robot.
> > >
> > > Responding to reviewer **zxcc**’s perhaps related concern about “insufficient method design”: our method was intended to be easy to implement within an existing RL setup. As stated by reviewer **Xsn5** _”The incorporation of the novel loss does not significantly enhance the complexity of the pipeline. These straightforward objectives maintain clarity and simplicity within the pipeline.”_ We maintain that the generalisability of our method is a strength.
> > >
> > > We believe that the above potentially signifies a misunderstanding from **zxcc** regarding our work. We can easily address this by updating the paper to emphasise these points in the introduction and conclusion.
> > >
> > > **Lack of real-world experiments**
> > >
> > > As we explained in our response [zxcc-7], we believe the absence of real-world experiments does not detract from the core contributions of this work. Our focus is on solving a fundamental machine learning problem in a simulated environment, which is a necessary first step. We submitted this work to a machine learning venue - NeurIPS - because we believe our primary contributions lie in the realm of machine learning methodology. We fully agree that real-world validation is important and view this as an exciting next step for future work, which we would pursue in a robotics-focused venue.
> > >
> > > We also wish to clarify that while two other reviewers mentioned the lack of real-world experiments, neither recommended rejection based on this. **gp4F**’s concern was “that the correlation between simulation experiments working versus the policy actually working on the real robot is quite large, especially for dexterous manipulation”, which we respond to in [gp4F-4] by referencing prior works that have successfully achieved this with nearly identical setups. **WBY5**’s concern was about the physical realism and transfer of superhuman policies, which we respond to in [WBY5-[1-3]].
> > >
> > >
> > > **Improvement to the manuscript text**
> > >
> > > As noted in our previous responses (see [zxcc-1,2,4]), we are planning on updating the methods section to improve clarity on the notation, Fig. 1 caption, explicitly provide the SSL + RL loss summation as an equation, and more clearly reference the implementation details provided in the Appendix. These are all easy changes to make. We highlight that other reviewers have commented on the quality of the manuscript: reviewer **gp4F** regarded it as “well-written” and reviewer **WBY5** scored the clarity as “excellent”.

---

### Note · Authors · 2025-08-14

We appreciate the thorough review process and the insightful feedback. Through the rebuttal and subsequent discussions, we have clarified several key points.

**Regarding concerns about realism**, our performance benefits are not due to exploiting non-realistic physics. We have demonstrated the same benefits across both static and dynamic tasks, which highlights that our agents learn genuinely superior policies rather than exploiting simulation inaccuracies (WBY5-1).

**For the sim-to-real transfer**, we are confident that the benefits of our method should translate to the real world, assuming the RL-only agent's sim-to-real gap can be successfully bridged (WBY5-3). However, not having these experiments does not detract from our contribution. We believe our comprehensive empirical study in controlled environments offers significant findings and novel insights. Establishing the efficacy of our method in a repeatable benchmark is a crucial first step for the community (gpF4-4).

**To further validate our hypotheses**, we will incorporate new experiments as discussed in [Xsn5-2]. Our initial results already provide strong evidence, and this additional validation will offer deeper insight into the unique challenges of this underexplored problem space.

The strong results, new environments, and open-source benchmark establish our work as a solid foundation that advances research in tactile-based RL for robotic control.

---

### Decision · Program_Chairs · 2025-09-17

**Decision:**

Accept (poster)

**Comment:**

This paper studies tactile manipulation and self-supervised approaches for it, creating simulation environments in Isaac Lab to benchmark different approaches.

For strengths, reviewers mentioned the finding of using self-supervised learning as representation learning as superior to "RL only", studying "tactile dynamics" is interesting, thorough hyperparameter and learning rate search, and that the peak performances are impressive.

For weaknesses, many reviewers mentioned the absence of sim2real experiments, a lack of technical contributions, that Fig 1. is very confusing, and missing discussion of some related work.

The method is not novel, but the paper claims that its application and analysis is (specializing model-based RL to tactile manipulation settings).

Overall, the claimed contributions are supported by evidence and the claimed contributions are interesting enough, despite the main limitation of absence of real robot evaluation. I remain on the fence about this paper due to the weaknesses.